# Topological effects and conformal invariance in long-range correlated random surfaces

Nina Javerzat[*], Sebastian Grijalva, Alberto Rosso and Raoul Santachiara

Université Paris-Saclay, CNRS, LPTMS, 91405, Orsay, France

[*] nina.javerzat@u-psud.fr

## Abstract

We consider discrete random fractal surfaces with negative Hurst exponent $H < 0$. A random colouring of the lattice is provided by activating the sites at which the surface height is greater than a given level $h$. The set of activated sites is usually denoted as the excursion set. The connected components of this set, the level clusters, define a one-parameter ($H$) family of percolation models with long-range correlation in the site occupation. The level clusters percolate at a finite value $h = h_c$ and for $H \leq -\frac{3}{4}$ the phase transition is expected to remain in the same universality class of the pure (i.e. uncorrelated) percolation. For $-\frac{3}{4} < H < 0$ instead, there is a line of critical points with continously varying exponents. The universality class of these points, in particular concerning the conformal invariance of the level clusters, is poorly understood. By combining the Conformal Field Theory and the numerical approach, we provide new insights on these phases. We focus on the connectivity function, defined as the probability that two sites belong to the same level cluster. In our simulations, the surfaces are defined on a lattice torus of size $M \times N$. We show that the topological effects on the connectivity function make manifest the conformal invariance for all the critical line $H < 0$. In particular, exploiting the anisotropy of the rectangular torus ($M \neq N$), we directly test the presence of the two components of the traceless stress-energy tensor. Moreover, we probe the spectrum and the structure constants of the underlying Conformal Field Theory. Finally, we observed that the corrections to the scaling clearly point out a breaking of integrability moving from the pure percolation point to the long-range correlated one.



# 1 Introduction

The percolative properties of random fractal surfaces have been studied for a long time [1–4]. The universality class of their critical points remains a very active subject of research in the mathematical [5–7] and in the theoretical physics [8] communities, mainly because they challenge our understanding of both the emergence of conformal symmetry and of the way this symmetry is implemented.

Let us consider a random stationary function $u(\mathbf{x})$ on a lattice $u(\mathbf{x}) : \mathbb{Z}^2 \to \mathbb{R}$ which satisfies:

$$\mathbb{E}\left[u(\mathbf{x})\right] = 0, \quad \mathbb{E}\left[(u(\mathbf{x}) - u(\mathbf{y}))^2\right] \sim C(H)\,|\mathbf{x} - \mathbf{y}|^{2H} \quad (|\mathbf{x} - \mathbf{y}| \gg 1), \tag{1}$$

where $\mathbb{E}\left[\cdots\right]$ is the average over the instances of $u(\mathbf{x})$, the symbol $\sim$ stands for asymptotically equivalent and $C(H)$ is some constant depending on $H$. The number $H$, $H \in \mathbb{R}$, is the surface roughness exponent [9], also known as Hurst exponent. The fractional Gaussian surfaces [7] that we consider here, see (4) below, is a class of random surfaces which satisfy the above properties. For positive $H > 0$, the function $u(\mathbf{x})$ is a fractional Brownian surface with unbounded height fluctuations, $\mathbb{E}\left[u(\mathbf{x})^2\right] = \infty$. The fluctuations remain unbounded also for $H = 0$ in which case the covariance decreases logarithmically, $\mathbb{E}\left[u(\mathbf{x})u(\mathbf{0})\right] \sim -\log|\mathbf{x}|$. For negative exponent $H < 0$, $u(\mathbf{x})$ is a long-ranged correlated surface with bounded fluctuations, $\mathbb{E}\left[u(\mathbf{x})^2\right] < \infty$.

A random partition of the lattice is obtained by setting a level $h$, $h \in \mathbb{R}$, and by declaring that a site $\mathbf{x}$ is activated (not activated) if $\theta_h(\mathbf{x}) = 1$ ($\theta_h(\mathbf{x}) = 0$), where $\theta_h(\mathbf{x}) : \mathbb{Z}^2 \to \{0, 1\}$:

$$\theta_h(\mathbf{x}) = \begin{cases} 0, & u(\mathbf{x}) < h \\ 1, & u(\mathbf{x}) \geq h. \end{cases} \tag{2}$$

A site is therefore activated with probability $p(h)$:

$$p(h) = \mathbb{E}\left[\theta_h(\mathbf{x})\right], \tag{3}$$

where we use the translational invariance in law. The set of activated points is usually known as the excursion set [10]. The study of the connected components of the excursion set, hereafter referred to as level clusters, defines a site percolation model [8, 11]. For general values of $H$ there is a finite value of $h = h_c > -\infty$ below which a level cluster of infinite size is found with probability one [12]. This is the percolation critical point. Note that the characterisation of the class of random fields which permit percolation has been given in [2, 13, 14]. Close to the critical point, the main scaling behaviours are described by two critical exponents, the correlation length $\nu$ and the order parameter $\beta$ exponents [11]. In particular, they determine the scaling of the $h_c$ width distribution with the size of the system, see (56), and the Hausdorff dimension $D_f$ of the level cluster, $D_f = 2 - \beta/\nu$. For $H > 0$, due to unbounded fluctuations of $u(\mathbf{x})$ and to the strong correlations, the level clusters are compact (i.e. without holes) regions with fractal dimension $D_f = 2$. The exponent $\nu$ is infinite $\nu = \infty$, as one can see from the fact that the $h_c$ width distribution remains finite in the thermodynamic limit (self-averaging is broken) [12]. At $H > 0$ the transition is not critical. At the point $H = 0$, the fluctuations of the surface remain unbounded and the fractal dimension remains $D_f = 2$, as argued in [15] and recently proven in [16,17] for the Gaussian free field. For negative roughness exponent instead, the surface fluctuations are bounded, the correlation length exponent $\nu$ is finite ($\nu < \infty$) and a genuine continous transition of percolation type occurs. Correspondingly, the level clusters have a richer fractal structure with $D_f < 2$.

In this paper we consider random surfaces with negative roughness exponent. If not stated otherwise, we take $H < 0$ henceforth. We generate a fractional Gaussian process on a flat torus of dimension $M \times N$. The surface $u(\mathbf{x})$ takes the form

$$u(\mathbf{x}) \propto \sum_{\mathbf{k}} \lambda_{\mathbf{k}}^{-\frac{H+1}{2}} \, \hat{w}(\mathbf{k}) \, e^{i\,\mathbf{k}\,\mathbf{x}}. \tag{4}$$

In the above equation $\lambda_{\mathbf{k}}$ and $e^{i\,\mathbf{k}\,\mathbf{x}}$ are respectively the eigenvalues and the eigenvectors of the discrete Laplacian operator $\Delta_{\mathbf{x}} u(\mathbf{x}) = \sum_{\mathbf{y},|\mathbf{y}-\mathbf{x}|=1} (u(\mathbf{y}) - u(\mathbf{x}))$ on the flat torus, and the $\hat{w}(\mathbf{k})$ are independent normally distributed random variables. The basic idea is to obtain correlated variables by convoluting uncorrelated ones. For $H = 0$ the function $u(\mathbf{x})$ is the discrete two-dimensional Gaussian free field on the torus. The role of open boundary conditions in one-dimensional fractional Gaussian processes is discussed in [18–20]. We generate also a second type of long range correlated random surface where the $\hat{w}(\mathbf{k})$ are drawn by a different distribution. Full details on how we generate the surfaces are given in Appendix A.

The probability of activating two distant sites inherits the long-range correlation of the random surfaces:

$$\mathbb{E}[\theta_h(\mathbf{x})\theta_h(\mathbf{y})] - p(h)^2 \sim C'(H)|\mathbf{x}-\mathbf{y}|^{2H} \quad (|\mathbf{x}-\mathbf{y}| \to \infty), \tag{5}$$

where $C'(H)$ is some constant depending on $H$ and on the chosen distribution. For $H = -1$ the surfaces we generate are an instance of the two-dimensional white noise where the probabilities of activating two different sites are uncorrelated ($C'(-1) = 0$ in the above equation). The point $H = -1$ corresponds therefore to the pure percolation point. In Figure 1 we show instances of the surfaces (4) and the corresponding excursion set and level clusters at the critical point.

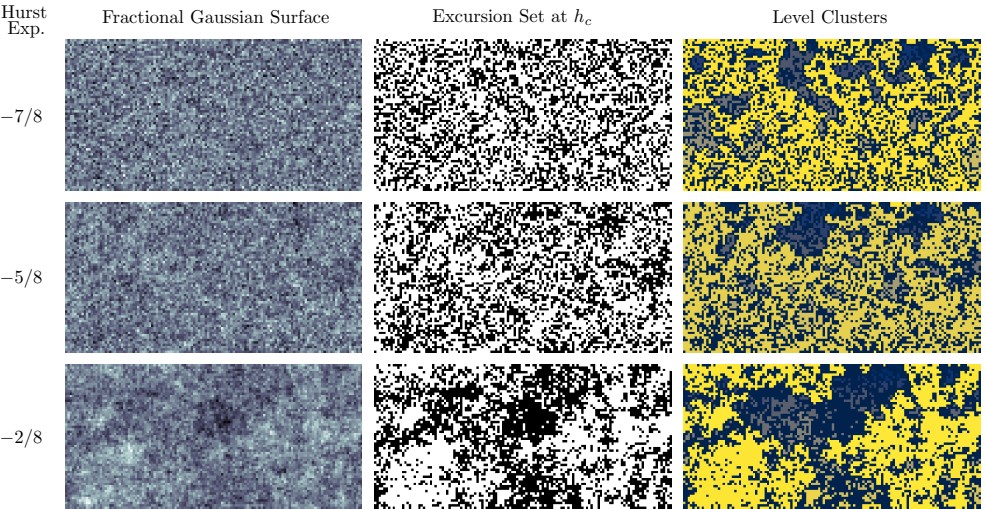

Figure 1: Instances of the fractional Gaussian surfaces (4) for $H \in \{-7/8, -5/8, -2/8\}$, generated on a $M \times N$ square lattice with $M = 2N$, $N = 2^6$. The excursion sets (white points) corresponding to level $h = h_c$ from Table 10 are shown in the second column, while the third column shows the level clusters. The yellow points in the third column are the points belonging to the percolating level cluster. Note that by increasing $H$, i.e. the correlation, the level clusters have less holes. This is consistent with the prediction that the fractal dimension $D_f \to 2$ for $H \to 0^-$.

The common understanding is that the percolating universal properties only depend on the asympotic behaviour of the covariance (1) and therefore on $H$. In [21] an extended Harris criterion was proposed, according to which the universality class remains the one of pure percolation for $H < -3/4$. Recent new arguments, based on the fractal dimension of the pivotal points support this prediction [22, 23]. The exponents $\nu$ and $D_f$ are expected to be

$$\nu = \nu^{\text{pure}} = \frac{4}{3}, \quad D_f = D_f^{\text{pure}} = \frac{91}{48}, \quad \text{for } H \leq -\frac{3}{4}, \tag{6}$$

where $\nu^{\text{pure}}$ and $D_f^{\text{pure}}$ are the pure percolation ($H = -1$) exponents. The fact that the system behaves as pure percolation for $H < -2$ was put on more rigorous grounds by [5, 24]. For $-3/4 < H < 0$ instead, the slower decay allows the correlation to change the large distance behaviour of the system, as was also argued in [3]. In particular, it was shown in [21] that there is a new line of critical points with an exponent $\nu = \nu^{\text{long}}$ which varies continuously with $H$:

$$\nu = \nu^{\text{long}} = -\frac{1}{H}, \quad -\frac{3}{4} < H < 0. \tag{7}$$

The above prediction was supported by many numerical works, see for instance [3, 4, 12, 25, 26]. There are no theoretical predition for $D_f$ in the range $-3/4 < H < 0$. In Figure 1 the level clusters become visibly more compact by increasing the value of $H$. One can expect then $D_f$ to increase when $H \to 0^-$. Even if the numerical results are not conclusive about the value of $D_f^{\text{long}}$, there are strong evidences that [12, 25–27]:

$$D_f = D_f^{\text{pure}} \quad \text{for } H \leq -\frac{1}{2}, \quad \text{and} \quad D_f^{\text{pure}} < D_f < 2 \quad \text{for } -\frac{1}{2} < H < 0. \tag{8}$$

In Appendix B, we numerically compute $D_f$. The results, summarised in Table 12, support the above scenario. The following diagram summarises the actual state of the art:

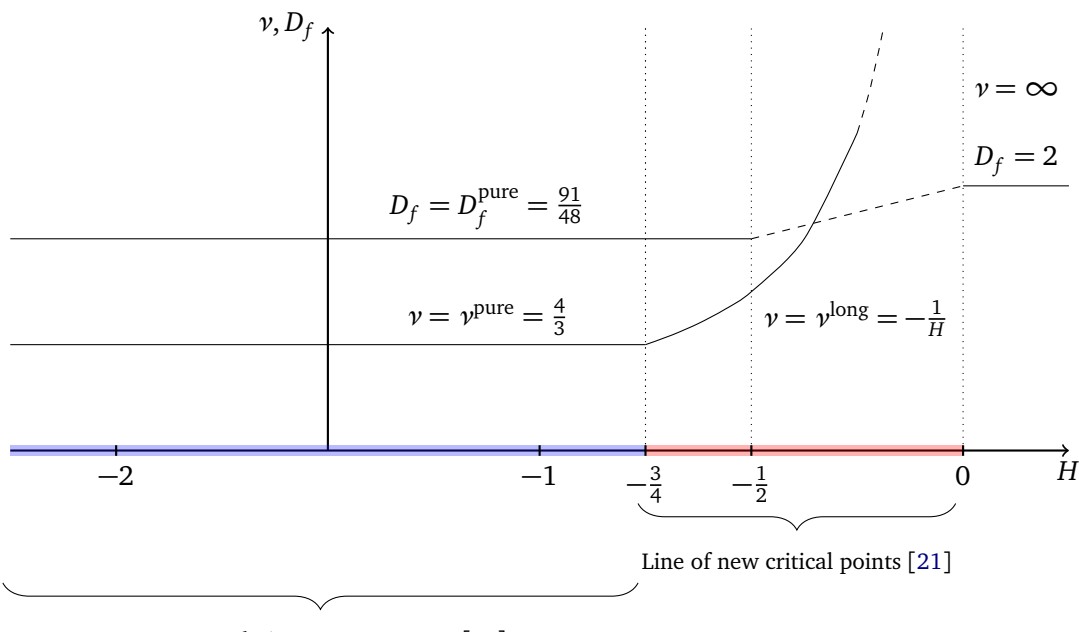

Figure 2

We stress the fact that the results mentioned above are based on the assumption that the kernel has a definite sign at large distances. For other important classes of random functions, this is not true anymore. This is the case for instance of the random plane wave [28]: this random function has an oscillating kernel which decays with an exponent $H = -1/4$, and the universality class of its percolation transition is conjectured to be the one of pure percolation [29].

Most of the results on critical pure percolation have been discovered by using conformal invariance [30]whose emergence has been rigourously proven in [31]. The values (6) have been predicted by the conformal field theory (CFT) approach [32,33], which allowed also the computation of the full partition function [34] and the derivation of exact formulas for cluster crossing probabilities [35]. Contrary to statistical models with local and positive Boltzmann weights, whose critical points are described by the unitary minimal models [36], the critical point of pure percolation is described by a non-unitary and logarithmic CFT [37,38]. This CFT is not fully known, but very recent results have paved the way to its complete solution [37–43]. The line of new critical points shown in Figure 2 remains by far less understood. As we will discuss below, even the emergence of conformal invariance is debated. Moreover, if these points are conformal invariant we expect that the corresponding CFT does not coincide with any of the known solutions, due to the highly non-local nature of the lattice model. This will be indeed confirmed by the results presented in this paper.

Recent numerical results have shown the emergence of conformal invariance [44], while in [45], where a random surface with $H = -2/3$ was considered, conformal symmetry has been ruled out. These papers check if the boundary of the percolation level cluster is described by a Stochastic Loewner Evolution (SLE) process [46]. The SLE numerical tests are in general very subtle and, in some cases not conclusive, as argued for instance in [47]. Moreover we observe that, in case of a positive SLE test as in [44], one expects the boundaries of the level clusters to be described by the loops of the $O(n)$ models either in their dense or critical phases [48]. In these models, the fractal dimensions of the loops $D_b$ and of their interior $D_f$ vary with $n$ [49]. For instance, in the $O(n)$ dense phase, they are related by $D_f = D_b(2-3D_b)/(4(1-D_b))$. This

scenario is not consistent with the numerical findings for the level clusters of long-range correlated random surfaces, as can be directly seen from the fact that $D_f$ does not show significant variation for $-3/4 < H < -1/2$ while $D_b$ does [44]. Moreover, we provide further evidences that the line $-3/4 < H < 0$ is not the one of the $O(n)$ models. This point illustrates the fact that many fundamental questions remain open.

Our objective is to test conformal invariance and to extract new information about these critical points. We use a completely different protocol based on the study of the level clusters and their connectivity function. This is the probability that two sites belong to the same level cluster, see (10). Because the random surfaces have double periodicity, the level clusters live on a torus. For pure percolation, signatures of conformal invariance were shown to be encoded in toric boundary conditions effects in the connectivity function [50]. These effects depend on a non trivial combination of the two exponents $\nu$ and $D_f$, fixed by conformal invariance. Moreover, when the lattice is rectangular, $M \neq N$, a soft breaking of rotational symmetry is introduced. Using this anisotropy, we show that the connectivity function directly probes the existence of the two components of a traceless stress-energy tensor. The existence of this pair of fields is the most basic manifestation of conformal symmetry. Finally, we provide the first numerical measurements of quantities related to the conformal spectrum and structure constants of this new conformal critical points.

In Section 2 we define the connectivity function and we give the theoretical predictions for the toric effects. We discuss the main ideas behind the CFT approach on which these predictions are based. In Section 3 we provide the numerical evidences on the connectivity function. In Appendix A we provide full details on how we generate the random surfaces and in Appendix B, on how we locate the critical percolation point and compute the exponents $\nu$ and $D_f$.

## 2 Critical two-point connectivity of level clusters

In this section we consider the two-point connectivity $p_{12}(\mathbf{x_1}, \mathbf{x_2})$, referred to as simply correlation function in [11]. Defining the event:

$$\text{Conn}(\mathbf{x_1}, \mathbf{x_2}) = \mathbf{x_1} \text{ and } \mathbf{x_2} \text{ belong to the same level cluster,} \tag{9}$$

we define:

$$p_{12}(\mathbf{x_1} - \mathbf{x_2}) = \mathbb{E}\left[\text{Conn}(\mathbf{x_1}, \mathbf{x_2})\right], \tag{10}$$

where translational invariance in law has been taken into account. A study of two-point connectivity for general Gaussian random surfaces can be found in [51] where the large $h$ asymptotic behaviour of (10) has been considered. Here we are interested in the behaviour of this probability at the critical point $h = h_c$.

### 2.1 Scaling limit in the infinite plane $M, N = \infty$

Let us consider first the regime in which toric size effects are negligeable. It corresponds to $M, N = \infty$, i.e. the infinite plane limit.

At the critical point, $h = h_c$, we have $p_{12}(\mathbf{x}) \sim |\mathbf{x}|^{-\eta}$, where $\eta$ is the standard notation for the anomalous dimension of the two-point function [11]. Percolation theory tells us that $\eta$ is directly related to the level cluster dimension $D_f$ via the scaling relation $\eta = 4 - 2D_f$ [52]. One has therefore:

$$p_{12}(\mathbf{x_1} - \mathbf{x_2}) = \frac{d_0}{|\mathbf{x_1} - \mathbf{x_2}|^{2(2 - D_f)}} \quad (|\mathbf{x_1} - \mathbf{x_2}| >> 1, \, M, N = \infty), \tag{11}$$

where $d_0$ is a non-universal constant which we evaluate numerically, see Table 1. We can use (11) to determine $D_f$. The corresponding values are denoted as $D_f^{(2)}$ in Table 12. The good agreement with the values $D_f^{(1)}$, obtained using the scaling of the average mass of the percolating level cluster (see Appendix B), confirms that we are sitting sufficiently close to the critical value $h_c$.

In Figure 3 we show the behaviour of $p_{12}(\mathbf{x_1} - \mathbf{x_2})$ for $H = -5/8$. One can easily notice a region $|\mathbf{x_1} - \mathbf{x_2}| \in [10, 100]$ where the form (11) is well satisfied.

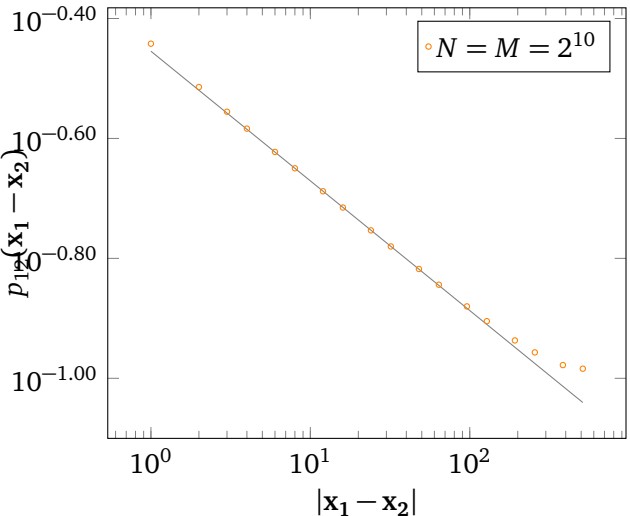

Figure 3: Two-point connectivity (10) for $H = -5/8$ and $N = M = 2^{10}$. The data points were obtained by averaging over $10^5$ instances of the surface and over the $N^2$ locations of point $\mathbf{x_1}$ (cf. Section 3). According to Table 10, the level $h$ has been set to $h_c = -0.1985$. The continuous line shows the prediction (11) with $D_f = D_f^{(2)} = 1.892$, see Table 12. For distances $6 < |\mathbf{x_1} - \mathbf{x_2}| < 100$ the data matches very well with the infinite plane prediction. For larger distances, the effect of the toric boundary conditions becomes visible.

## 2.2 Scaling limit in the torus: $M, N < \infty$.

As can be seen in Figure 3, when the distance between points approaches $N/2$, the data points start to deviate from the power-law behaviour: the contributions of the paths connecting the two points on the other side of the torus become non negligeable. We say that the topological corrections become visible. We expect these corrections to provide sub-leading $|\mathbf{x}|/N$ terms in (11) of universal nature. These effects have been studied in [50] for pure percolation ($H = -1$).

In the scaling limit, our system lives on a flat torus $\mathbb{T}_q$ of periods $M$ and $N$ and nome $q$:

$$\mathbb{T}_q : \quad q = e^{-2\pi \frac{M}{N}}. \tag{12}$$

As the connectivity between two points always depend on the vector connecting them, it is convenient to introduce the vector $\mathbf{x}, \mathbf{x}^\perp \in \mathbb{T}_q$ that have polar coordinates $|\mathbf{x}|$ and $\theta$:

$$\mathbf{x} \in \mathbb{T}_q, \quad \mathbf{x} = |\mathbf{x}|(\cos(\theta), \sin(\theta)), \quad \mathbf{x}^\perp = |\mathbf{x}|(-\sin(\theta), \cos(\theta)). \tag{13}$$

We propose the following form for the scaling limit of $p_{12}$ on a torus:

$$p_{12}(\mathbf{x}) = \frac{d_0}{|\mathbf{x}|^{2(2-D_f)}} \left( 1 + c_\nu(q) \left( \frac{|\mathbf{x}|}{N} \right)^{2 - \frac{1}{\nu}} + 2c_T(q) \cos(2\theta) \left( \frac{|\mathbf{x}|}{N} \right)^2 + o\left( \left( \frac{|\mathbf{x}|}{N} \right)^2 \right) \right), \tag{14}$$

which has been established in [50] for pure percolation and for the more general random cluster $Q$−Potts model. The coefficients $c_v(q)$, and $c_T(q)$, given in (19), are universal coefficients which depend only on the geometry of the torus. To explain the origin of (14) and the information we can extract from this formula, we need to recall some basic definitions and notions on CFT.

## 2.3 Basic notions of CFT

A CFT is a massless quantum field theory in which each (quantum) field $V_{\Delta,\bar{\Delta}}(\mathbf{x})$ is characterised by a pair of numbers $(\Delta, \bar{\Delta})$, called left and right conformal dimensions, which give the scaling dimension ($\Delta^{\text{phys}} = \Delta + \bar{\Delta}$) and the spin ($s = \Delta - \bar{\Delta}$) of the field. The set of fields entering a CFT is called the spectrum $\mathcal{S}$ of the theory, $\mathcal{S} = \oplus_{(\Delta,\bar{\Delta})} V_{(\Delta,\bar{\Delta})}$. The most important landmark of conformal invariance is the existence of two fields, commonly denoted as $T$ and $\bar{T}$, with left-right dimensions $(\Delta, \bar{\Delta}) = (2, 0)$ and $(\Delta, \bar{\Delta}) = (0, 2)$. These fields are the conserved (chiral) Noether current associated to the conformal symmetry, and they correspond to the components of the traceless stress-energy tensor field.

In the CFT approach to statistical models, there is a correspondence between lattice operators and fields $V_{\Delta,\bar{\Delta}}(\mathbf{x})$. In particular, the long distance behaviour of lattice observables is described by the correlation functions of the fields $V_{\Delta,\bar{\Delta}}(\mathbf{x})$. Scale invariance fixes the infinite plane limit of the two-point functions. For a spinless field $V_{\Delta,\Delta}$ we have:

$$\left\langle V_{\Delta,\Delta}(\mathbf{x}) V_{\Delta,\Delta}(0) \right\rangle_q = |\mathbf{x}|^{-4\Delta} \quad \left( \frac{|\mathbf{x}|}{N} \to 0 \right), \tag{15}$$

where $\langle \cdots \rangle_q$ denotes the torus CFT correlation function on $\mathbb{T}_q$. A quantum field theory is completely solved if we can compute all its correlation functions. For a CFT, one needs two basic inputs: the spectrum $\mathcal{S}$ and the structure constants $C_{V_1,V_2}^{V_3}$. The latter are real constants associated to the amplitude with which two fields $V_1$ and $V_2$ fuse into a third one $V_3$. Said in other words, the constants $C_{V_1,V_2}^{V_3}$ determine the short-distance behaviour of the CFT correlation functions which is encoded, in the CFT jargon, in the Operator Product Expansion (OPE).

Among all the fields in a CFT, a major role is played by the density energy field $\varepsilon = V_{\Delta_\varepsilon, \Delta_\varepsilon}$ and the magnetic (order parameter) field $\sigma = V_{\Delta_\sigma, \Delta_\sigma}$, which are the (spinless) fields with the lowest scaling dimension in the thermal and magnetic sector. Their names come from the fact that, in a ferromagnetic/paramagnetic type transition, these are the fields which couple respectively to the temperature and to the magnetic field. Their dimensions $\Delta_\varepsilon$ and $\Delta_\sigma$ give the exponents $\nu$ and $\beta$ of a critical point [53, Chapter 3]. In terms of $\nu$ and $D_f = (4-\eta)/2 = 2-\beta/\nu$ [11, Section 3.3] we have:

$$\Delta_\varepsilon = 1 - \frac{1}{2\nu}, \quad \Delta_\sigma = 1 - \frac{D_f}{2}. \tag{16}$$

## 2.4 Three main assumptions

Our prediction (14) is based on three assumptions which have been verified for pure percolation [50, 54]. The first two assumptions are more general and concern the fact that the connectivity, which is non-local in nature, can be studied by correlations of local fields in a CFT.

- **1:** The system is conformally invariant in the scaling limit.

- **2:** The scaling limit of the connectivity (10) is described by the two spin field torus correlator:

$$p_{12}(\mathbf{x}) = d_0 \, \langle \sigma(\mathbf{x}) \sigma(0) \rangle_q. \tag{17}$$

The two-point function $\langle \sigma \sigma \rangle_q$ can be expressed as an ($s$-channel) expansion:

$$
\begin{aligned}
p_{12}(\mathbf{x}) &= d_0 \langle \sigma(\mathbf{x}) \sigma(0) \rangle_q \\
&= \frac{d_0}{|\mathbf{x}|^{4\Delta_\sigma}} \sum_{\substack{V_{\Delta,\bar{\Delta}} \in \mathcal{S} \\ \Delta \geq \bar{\Delta}}} (2 - \delta_{\Delta,\bar{\Delta}}) C_{\sigma,\sigma}^{V_{\Delta,\bar{\Delta}}} \langle V_{\Delta,\bar{\Delta}} \rangle_q \cos\left((\Delta - \bar{\Delta})\theta\right) \left(\frac{|\mathbf{x}|}{N}\right)^{\Delta+\bar{\Delta}},
\end{aligned}
\tag{18}
$$

with $\mathbf{x} = |\mathbf{x}|(\cos(\theta), \sin(\theta))$, see (13). In general, $p_{12}$ does not get contributions from all the fields in the spectrum $\mathcal{S}$, since structure constants $C_{\sigma,\sigma}^{V_{\Delta,\bar{\Delta}}}$ and/or one-point functions $\langle V_{\Delta,\bar{\Delta}} \rangle_q$ may vanish. We refer the reader to [50] for a detailed derivation of the above formula which is a direct consequence of the existence of an operator algebra and of the symmetry between the holomorphic and anti-holomorphic sectors. This latter symmetry is very natural for CFTs without boundaries and implies that if a field with spin $s > 0$ enters in the spectrum, then also its anti-holomorphic partner does, with the same physical dimension and with spin with opposite sign $-s$. The expansion (18) is then valid for almost all the CFTs. The information which characterise a specific CFT is encoded in the spectrum $\mathcal{S}$ and in the structure constants $C_{\sigma,\sigma}^{V_{\Delta,\bar{\Delta}}}$. In the case of pure percolation, for instance, the spectrum is known but not the structure constants, even if very recent progresses have paved the way to their determination [43]. The plane limit $M, N = \infty$ is recovered by noting that all the one-point functions $\langle V_{\Delta,\bar{\Delta}} \rangle_q$ vanish but the identity one $\langle \mathrm{Id} \rangle_q = \langle V_{0,0} \rangle_q = 1$. One obtains $p_{12}(\mathbf{x}) = d_0 |\mathbf{x}|^{-4\Delta_\sigma}$ ($M, N = \infty$). Note that, in the infinite plane limit, one can prove for pure percolation (or more generally for the $O(n)$ models in their dense or critical phases) that $p_{12}$ is given by the correlator of two spin fields $\sigma$, see for instance [49, 55]. The exponent $\eta$ is therefore $\eta = 4\Delta_\sigma$ which, by (16) gives equation (11).

It has been shown in [50] that the first dominant terms in the above series can be computed for pure percolation. Our third assumption is motivated by a generalisation of these results to the case of long-range percolation:

- **3:** The identity field ($\Delta = \bar{\Delta} = 0$), the density energy density field $\varepsilon$ and the stress-energy tensor fields $T$ ($\Delta = 2, \bar{\Delta} = 0$), $\bar{T}$ ($\Delta = 0, \bar{\Delta} = 2$) are the fields with the lowest conformal dimension that appear in the fusion of two fields $\sigma$ and whose torus one-point function does not vanish.

Using the above assumption in the expansion (18), one obtains expression (14) with the coefficients $c_\nu(q)$ and $c_T(q)$ given by:

$$
c_\nu(q) = C_{\sigma,\sigma}^\varepsilon \langle \varepsilon \rangle_q, \quad c_T(q) = C_{\sigma,\sigma}^T \langle T \rangle_q = \frac{2\Delta_\sigma}{c} \langle T \rangle_q,
\tag{19}
$$

where $c$ is the CFT central charge (which provides for instance the universal Casimir amplitude [56]). We refer the reader to [50, 54] for a detailed explication of the CFT techniques used to study the topological effects.

Let us detail further the information one can extract from $c_\nu(q)$ and $c_T(q)$. The spectrum $\mathcal{S}$ and some structure constants $C_{V_1,V_2}^{V_3}$ enter in the determination of these coefficients. For a general CFT, the spectrum defines the torus partition function [57]:

$$
Z(q) = q^{-\frac{c}{12}} \sum_{V_{\Delta,\bar{\Delta}} \in \mathcal{S}} n_{V_{\Delta,\bar{\Delta}}} q^{\Delta+\bar{\Delta}},
\tag{20}
$$

where $n_{V_{\Delta,\bar{\Delta}}}$ is the multiplicity of the field $V_{\Delta,\bar{\Delta}}$. For small values of $q$, the leading contributions to the partition function are given by the representations with the smallest physical dimensions. The Identity field $V_{0,0}$ has the lowest physical dimension 0, with $n_{\mathrm{Id}} = 1$. We will assume that

the sub-leading contribution to the partition function is given by a spinless field $V_{\Delta,\Delta}$ with multiplicity $n_{V_{\Delta,\Delta}}$. For non unitary CFTs, the number $n_{V_{\Delta,\Delta}}$ can take general real values. This is the case of the $Q-$ state Potts model [34], in which the sub-dominant contribution is given by the spin field $\sigma$ with multiplicity $n_\sigma = Q-1$.

In a general CFT, one-point torus functions can be expressed in the variable $q$, in a way similar to the partition function (20). As detailed in [50], the three assumptions of Section 2.4 lead to the following form for the energy density one-point torus function:

$$\langle \varepsilon \rangle_q = \frac{(2\pi)^{2\Delta_\varepsilon}}{Z(q)} C^\varepsilon_{\sigma,\sigma}\, n_\sigma q^{2\Delta_\sigma - \frac{c}{12}}\left(1 + O(q)\right). \tag{21}$$

The coefficient $c_\nu(q)$, given by (19), can therefore be expanded in $q$ as:

$$c_\nu(q) = (2\pi)^{2\Delta_\varepsilon}\left[C^\varepsilon_{\sigma,\sigma}\right]^2 n_\sigma q^{2\Delta_\sigma} + o(q^{2\Delta_\sigma}). \tag{22}$$

In a similar way, using the formula [57]:

$$\langle T \rangle_q = -(2\pi)^2 q\, \partial_q \ln Z(q), \tag{23}$$

and expression (20) of the partition function, the coefficient $c_T(q)$ (given by (19)) admits the following small $q$ expansion:

$$c_T(q) = \frac{(2-D_f)\pi^2}{6}\left(1 - 24\Delta\frac{n_{V_{\Delta,\Delta}}}{c}q^{2\Delta} + \cdots\right). \tag{24}$$

The above three assumptions do not put any constraint on the dimension $\Delta$ and multiplicity $n_{V_{\Delta,\Delta}}$ of the field giving the leading contribution to (24). For pure percolation, for which the partition function (20) is known exactly, this leading contribution is given by the spin field $\sigma$:

$$c_T(q) = \frac{(2-D_f)\pi^2}{6}\left(1 - 12(2-D_f)\frac{n_\sigma}{c}q^{2-D_f} + \cdots\right). \tag{25}$$

In that case the ratio $n_\sigma/c$ can be obtained as the limit $Q \to 1$ of the analogous expression for the $Q-$ Potts model. Using the fact that in this limit the central charge $c_Q \sim Q-1$ ($|Q-1| \ll 1$), the limit $c \to 0$ of $n_\sigma/c$ yields a finite non-zero limit, $n_\sigma/c = 4\pi/(5\sqrt{3})$.

## 2.5 Numerical protocols for testing CFT predictions

We have seen that, by using a CFT approach, the topological effects on $p_{12}$ encode in principle highly non-trivial information about the critical point. We discuss now how to efficiently extract this information from a numerical study of $p_{12}$ and how to interpret these results.

The torus shape can be exploited to disentangle the contributions of sub-leading and sub-sub leading terms in (14). This can be done by comparing the connectivities $p_{12}(\mathbf{x_2} - \mathbf{x_1})$ and $p_{12}(\mathbf{x_3} - \mathbf{x_1})$ between pairs of points $\mathbf{x_2}$ and $\mathbf{x_1}$ and $\mathbf{x_3}$ and $\mathbf{x_1}$ that are aligned on orthogonal axes, as illustrated in Figure 4. Note that similar ideas were used in [50].

Let us consider first the square torus, $M = N$ or $q = e^{-2\pi}$ and the case where $\mathbf{x_2} - \mathbf{x_1} = \mathbf{x}^h$ and $\mathbf{x_3} - \mathbf{x_1} = \mathbf{x}^v$ with $\mathbf{x}^h = |\mathbf{x}|(1,0)$ and $\mathbf{x}^v = |\mathbf{x}|(0,1)$. As the two cycles are equivalent, one has $p_{12}(\mathbf{x}^h) = p_{12}(\mathbf{x}^v)$. From (14) and (18), $p_{12}(\mathbf{x}^h) - p_{12}(\mathbf{x}^v) \sim 4\sum_{\Delta-\bar\Delta\neq0} C^{V_{\Delta,\bar\Delta}}_{\sigma,\sigma}\left\langle V_{\Delta,\bar\Delta}\right\rangle_{q=e^{-2\pi}} N^{-\Delta-\bar\Delta}$, which implies $\left\langle V_{\Delta,\bar\Delta}\right\rangle_{q=e^{-2\pi}} = 0$ if $\Delta - \bar\Delta \neq 0$. In particular $\langle T \rangle_{q=e^{-2\pi}} = 0$ and therefore:

$$c_T(e^{-2\pi}) = 0. \tag{26}$$

The connectivity (14) therefore reduces to:

$$p_{12}(\mathbf{x}) = \frac{d_0}{|\mathbf{x}|^{2(2-D_f)}} \left( 1 + c_\nu(q) \left( \frac{|\mathbf{x}|}{N} \right)^{2-\frac{1}{\nu}} + o\left( \left( \frac{|\mathbf{x}|}{N} \right)^2 \right) \right), \quad \text{for } M = N. \tag{27}$$

Let us consider now the rectangular torus $M > N$ with again $\mathbf{x_2} - \mathbf{x_1} = \mathbf{x}^h$ and $\mathbf{x_3} - \mathbf{x_1} = \mathbf{x}^v$. In Figure 5 we show the corresponding measurements of $p_{12}(\mathbf{x}^h)$ and $p_{12}(\mathbf{x}^v)$ when $M = 2N$. The two connectivities are now different, which is explained by the simple fact that the paths closing on the other side of the small cycle $(N)$ start to contribute for smaller distances than the ones closing on the largest one $(M)$. From (14) and for general $\mathbf{x}$ we have:

$$p_{12}(\mathbf{x}) - p_{12}(\mathbf{x}^\perp) = \frac{d_0}{|\mathbf{x}|^{2(2-D_f)}} \left( 4\cos(2\theta) \frac{2\Delta_\sigma}{c} \langle T \rangle_q \left( \frac{|\mathbf{x}|}{N} \right)^2 + o\left( \left( \frac{|\mathbf{x}|}{N} \right)^2 \right) \right), \tag{28}$$

where $\mathbf{x}$ and $\mathbf{x}^\perp$ are parametrised as in (13), and $c_T(q)$ has been replaced by its expression (19).

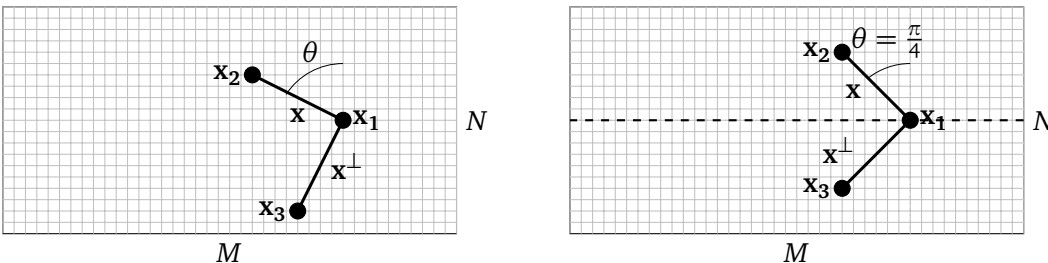

Figure 4: Left: We take three points $\mathbf{x_1}, \mathbf{x_2}, \mathbf{x_3}$ on the torus lattice $\mathbb{Z}^2/(N\mathbb{Z} \times M\mathbb{Z})$ such that $\mathbf{x_2} - \mathbf{x_1} = \mathbf{x}$ and $\mathbf{x_3} - \mathbf{x_1} = \mathbf{x}^\perp$, see (13). We measure $p_{12}(\mathbf{x})$ and $p_{12}(\mathbf{x}^\perp)$, defined in (10). Right: When $\theta = \pi/4$, $\mathbf{x}$ and $\mathbf{x}^\perp$ are symmetric by reflection with respect to the axis parallel to the $M$ axis and passing through $\mathbf{x_1}$ (dashed line). This implies $p_{12}(\mathbf{x}) = p_{12}(\mathbf{x}^\perp)$ for $\theta = \pi/4$.

Equation (28) is a clear consequence of the fact that, whenever an anisotropy is introduced, the response of the system is bound to be determined by the stress-energy tensor components $T$ and $\bar{T}$ (see for instance Section 11.3 in [53]). It is interesting to note that Monte Carlo algorithms, based on the properties of rectangular torii [58,59], have been proposed to measure the central charge and the leading fields in the partition function [60]. However, these methods can be applied to statistical models for which a direct lattice representation of the stress-energy tensor is available, such as the Ising model or the RSOS models [61]. In our case we do not know the stress-energy lattice representation. Actually, away from the pure percolation point $H = -1$, we do not even know the energy density lattice representation. This is also the reason why the connectivity functions are the most natural observables to study universal critical amplitudes of non-local models. Note that other non-scalar observables have been defined and discussed in [62,63], where the angular dependence of their two-point function has been measured by Monte-Carlo simulations.

From the expansion (18) of the connectivity, the difference (28) gets in general contributions only from fields with a non-zero spin. By lattice symmetry arguments, this difference vanishes for $\theta = \pi/4$, as shown in Figure 4. One can directly see from (18) that the only fields which may contribute to (28) are fields with spin $\Delta - \bar{\Delta} = 2 \bmod 4$. For instance one expects in (28) a contribution from fields with $(\Delta, \bar{\Delta}) = (6, 0)$ and $(\Delta, \bar{\Delta}) = (4, 2)$. These fields exist in any CFT as, said in CFT jargon, they correspond to the higher level descendants of the identity: $L_{-6}V_{0,0}$, $L_{-4}L_{-2}V_{0,0}$ and $L_{-4}\bar{L}_{-2}$, $L_{-2}^2\bar{L}_{-2}V_{0,0}$. In pure percolation there are no fields in the

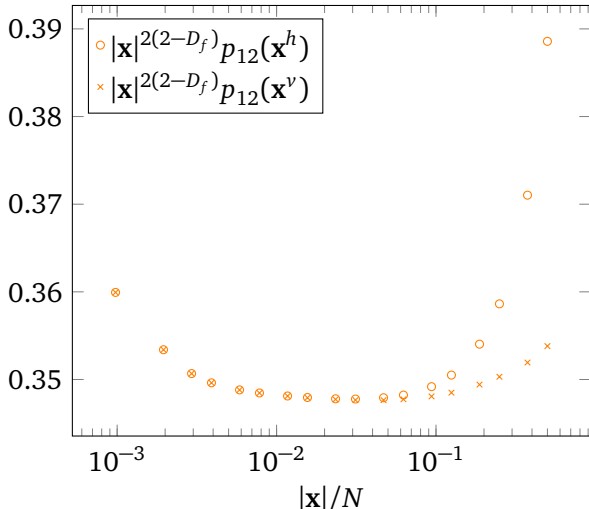

Figure 5: The connectivity measured for $H = -2/3$, along the small cycle (circles) and the long cycle (crosses) of a torus with $M/N = 2$, $N = 2^{10}$. The data points were obtained by averaging over $10^5$ instances of the surface and over the $N \times M$ locations of $\mathbf{x}$ (cf. Section 3. The connectivity measured along the long cycle of the torus is always smaller than the connectivity measured along the small cycle.

spectrum with spin greater than 2 and physical dimension $\Delta + \bar{\Delta} < 6$. If we assume this is true also for correlated percolation $H > -1$, then we have:

$$
\begin{aligned}
p_{12}(\mathbf{x}) - p_{12}(\mathbf{x}^\perp) = \frac{d_0}{|\mathbf{x}|^{2(2-D_f)}} \Bigg( & 4\cos(2\theta)c_T(q)\left(\frac{|\mathbf{x}|}{N}\right)^2 \\
& + 4\big[\cos(2\theta)c_{6,2}(q) + \cos(6\theta)c_{6,6}(q)\big]\left(\frac{|\mathbf{x}|}{N}\right)^6 + o\left(\left(\frac{\mathbf{x}}{N}\right)^6\right) \Bigg).
\end{aligned}
\tag{29}
$$

Assuming that the identity descendants are the only fields contributing to $c_{6,2}$ and $c_{6,6}$, these coefficients can be fixed by computing the inner products and the matrix elements between the 11 identity descendants existing at level 6. We refer the reader to [50, 54] and references therein for the details of the general procedure. However, the numerical determination of these coefficients is not accurate enough for this cumbersome computation to be worth it. As a matter of fact we use this order 6 term as a fitting parameter to obtain better estimations of the order 2 coefficient.

## 2.6 Numerical evidences

We summarise here the main numerical results for $p_{12}$ and the conclusions we can draw by comparing these results with the CFT predictions.

### 2.6.1 Conformal invariance

The quantity (14) is, first of all, a powerful test of conformal invariance. Via the numerical simulation of the connectivity we test two predictions:

- The dominant topological correction shows a precise interplay between the exponents $\nu$ and $D_f$. In particular the leading correction behaves as $|\mathbf{x}|^{2(2D_f-2)}(|\mathbf{x}|/N)^{2-1/\nu}$. This effect is more clearly seen on the square torus, see (27). Figure 9 shows that the numerical results for the values $H < -1/2$ agree with this prediction.

- The sub-leading term is $\propto |\mathbf{x}|^{2(2D_f-2)} \cos(2\theta)(|\mathbf{x}|/N)^2$. As explained above, the presence of such term implies the existence of a pair of fields with scaling dimension $\Delta + \bar{\Delta} = 2$, which corresponds to the power 2 in the $(|\mathbf{x}|/N)^2$ decay, and with spin $\Delta - \bar{\Delta} = \pm 2$, which fixes the $\theta$ dependence. If such fields exist, they correspond by definition to the stress-energy tensor components $T$ and $\bar{T}$. The presence of $T$ and $\bar{T}$ is the most basic and direct consequence of conformal invariance. In numerical simulations, the sub-leading term is seen by considering a rectangular torus. Figures 10 and 11 show clearly the $(|\mathbf{x}|/N)^2$ decay and the $\cos(2\theta)$ dependence. Figure 12 shows further that the data is well described by the form (29).

### 2.6.2   Spectrum and structure constants

- The values of $c_\nu(q)$ for different values of $q$ have been measured for $-1 < H < -1/2$ and reported in Table 2. The results support the fact that for $H \leq -3/4$ the universality class is the one of pure percolation. Note that this a highly non-trivial verification, as it not only based on the values of critical exponents, but on the values of constants which depend on the spectrum and fusion coefficients of the theory. For $H > -3/4$, the data are quite well consistent with the CFT prediction (21), as shown in Figure 13. This is also consistent with the fact that the fusion between two spin field produces an energy field.

- We could measure with good precision the dependence of the coefficient $c_T(q)$ with $q$. Figure 15 shows that (25) is satisfied, and that the dimension of the most dominant field coincides with the dimension of the spin field.

## 3   Numerical results on two point connectivity

We generate the random surfaces (48, 52) and we measure the connectivity (10) of its level clusters, for the following set of values of $H$:

$$H = -\frac{7}{8}, -\frac{2}{3}, -\frac{5}{8}, -\frac{21}{40}, -\frac{19}{40}, -\frac{3}{8}, -\frac{1}{4} \tag{30}$$

which are representative for the line $-1 \leq H < 0$. Due to the periodicity properties (55), we have a site percolation model on a doubly-periodic lattice of size $M \times N$, i.e. the toric lattice $\mathbb{Z}^2/(N\mathbb{Z} + M\mathbb{Z})$. In the square torus case ($M = N$), $p_{12}(\mathbf{x}) = p_{12}(|\mathbf{x}|)$. Without losing generality we measure $p_{12}$ between pairs of points $\mathbf{x_1}$ and $\mathbf{x_2}$, aligned on the vertical or horizontal axes. For each $H$ in (30), the data are taken for distances $|\mathbf{x_1} - \mathbf{x_2}| = |\mathbf{x}| = 1, 2, 4, \cdots, N/2$, $|\mathbf{x}| = 3, 6, 12, \cdots, 3N/8$. For the rectangular torus, $M \neq N$, we measure the connectivity between the points $\mathbf{x_1}$ and $\mathbf{x_2}$, and between $\mathbf{x_1}$ and $\mathbf{x_3}$, $\mathbf{x_3} - \mathbf{x_1} = (\mathbf{x_2} - \mathbf{x_1})^\perp = \mathbf{x}^\perp$, see Figure 4. When $\mathbf{x}$ and $\mathbf{x}^\perp$ are aligned with the cycles of the torus ($\theta = 0$), measurements are taken for aspect ratios $M/N = 1, 2 \cdots 5$, and for distances $|\mathbf{x}| = 1, 2, 4, \cdots, N/2$, and $|\mathbf{x}| = 3, 6, 12, \cdots, 3N/8$. Fixing the aspect ratio, we measured $p_{12}(\mathbf{x})$ for non-zero angles $\theta$. On the lattice, angles are of the form $\theta = \arctan\left(\frac{a_2}{a_1}\right)$, with $a_2$ (resp.$a_1$) a given number of lattice sites in the $M$ (resp.$N$) direction. Distances are then taken to be $|\mathbf{x}| = \sqrt{a_1^2 + a_2^2}(1, 2, 4, \cdots)$, $|\mathbf{x}| = \sqrt{a_1^2 + a_2^2}(3, 6, 12, \cdots)$, such that $|\mathbf{x}| \leq N/2$. We chose angles $\theta = 0, \arctan(1/4)$, $\arctan(1/3), \arctan(1/2), \arctan(2/3)$, for fixed aspect ratio $M/N = 3$.

Exploiting the translational invariance of the surface distribution, we average over the position $\mathbf{x_1}$ for each instance of $u(\mathbf{x})$, and then over $10^5$ instances. In the scaling limit, the dependence of $p_{12}(\mathbf{x})$ with respect to the lattice size $N$ is expected to be of the form $|\mathbf{x}|/N$. Plotting the

connectivity as a function of $|\mathbf{x}|/N$, we observe that the corrections to the scaling are still visible as the data points for different sizes do not collapse at large distances. In Figure 6a we show the data for $H = -5/8$ and for lattice sizes $M = N = 2^9 - 2^{12}$. One can see that the scaling limit is still not attained. These non-universal effects become even more important for larger $H$. As shown in Figure 6b for $H = -3/8$, even the infinite plane scaling limit is not clearly attained at the sizes of our simulations. Of course these non-universal effects make the analysis of the universal topological effects less precise, in particular for studying the contributions of the spinless fields. On the other hand, we observed that the non-universal effects are less important for the surface (52) generated by the kernel $\hat{S}_2(\mathbf{k})$, at least for values of $H < -1/2$. This is shown in 7b. For values of $H < -1/2$ and for the two surfaces (48) and (52) we could determine the non-universal constant $d_0$, as well as the dimension of the leading spinless contribution. For this latter, the consistency of the results obtained from the two surfaces makes the verification of the CFT predictions more solid. The coefficient $c_v$ and its dependence on the aspect ratio, on the other hand, could only be determined with sufficient precision for the surface (52).

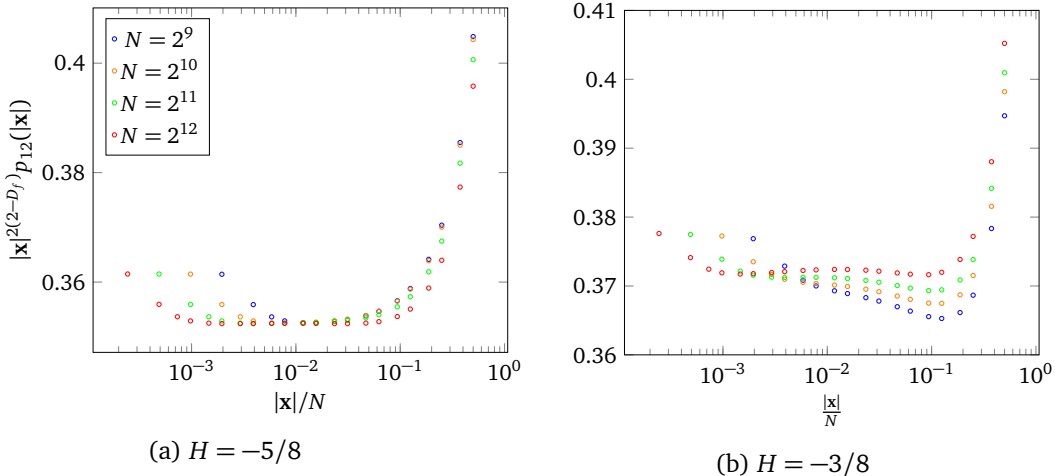

(a) $H = -5/8$

(b) $H = -3/8$

Figure 6: Convergence of the data points generated with surface (48), on the square torus of different sizes, for $H = -5/8$ (a) and $H = -3/8$ (b). Error bars are smaller than the marker size and we do not display them.

A very remarkable fact is that, for both surfaces, these corrections to the scaling terms cancel when one takes the differences between connectivities. This is shown in Figure 8 for the same values of $H$. The corrections may originate, for instance, from the fact that we are not sufficiently close to the critical point. More generally, any perturbation that drives the system out of the critical point and that does not break rotational invariance is related to a spinless field, whose contributions to the connectivity are isotropic. This explains why they disappear by taking the difference $p_{12}(\mathbf{x}) - p_{12}(\mathbf{x}^\perp)$. This mechanism allows to test the contribution of the fields with spin, and therefore of the stress-energy tensor, with a very good precision. For $H < -1/2$, our determination of the constants $d_0$ allowed moreover to acces the value of the universal coefficient $c_T(q)$. For $H > -1/2$, we could only determine the behaviour of $d_0 c_T(q)$ with $q$.

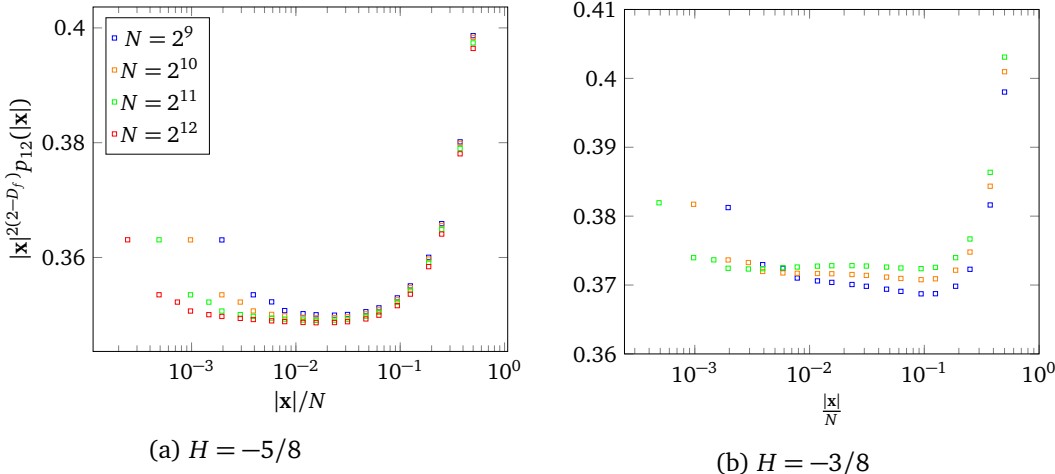

(a) $H = -5/8$

(b) $H = -3/8$

Figure 7: Convergence of the data points generated using the surface (52), on the square torus of different sizes, for $H < -5/8$ (a) and $H = -3/8$ (b).

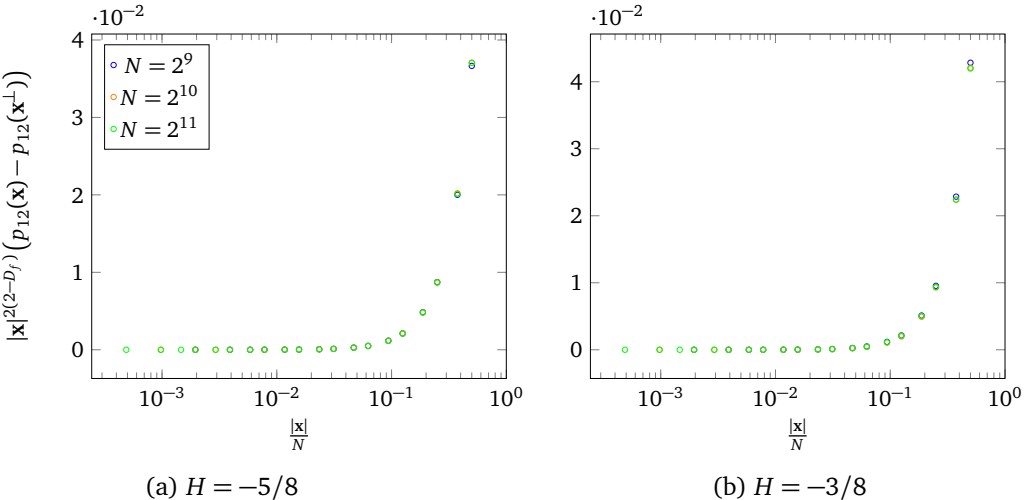

(a) $H = -5/8$

(b) $H = -3/8$

Figure 8: Convergence of the data points for the difference of connectivities (28) on rectangular torus $M = 2N$, for $H = -5/8$ (a) and $H = -3/8$ (b).

## 3.1 Plane limit

For $N = M = 2^{12}$, we fit the data points for $|\mathbf{x}| \in [4, 128]$, expected to be well described by the infinite plane limit (11) (see Figure 3), to the form

$$p_{12}(\mathbf{x}) \sim |\mathbf{x}|^{-2(2-D_f^{(2)})}. \tag{31}$$

The values $D_f^{(2)}$ of the fractal dimension are given in Table 12. To extract the topological corrections (27), we fit our numerical data to the form:

$$|\mathbf{x}|^{2(2-D_f^{(2)})} p_{12}(r) = d_0 \left(1 + \frac{d_1}{|\mathbf{x}|^{b_1}}\right)\left(1 + c_\nu \left(\frac{|\mathbf{x}|}{N}\right)^{2-1/\nu}\right). \tag{32}$$

The first factor takes into account the non-universal, small distance effects due to the lattice. We refer the reader to [39,41] for a more detailed discussion of these ultraviolet corrections. The values of $d_0$ are reported in Table 1. The numerical values for the universal coefficient $c_\nu$

are given in Table 2. They were obtained from the data generated using kernel (50), which converge faster to the scaling limit, and for which the agreement with (32) is excellent. This is shown in Figure 9.

Table 1: Non universal constant $d_0$ determined from the fit (32), for surfaces generated (1) with kernel (46) and (2) with kernel (50).

| $H$ | $d_0^{(1)}$ | $d_0^{(2)}$ |
|---|---|---|
| -7/8 | 0.3438(1) | 0.3433(2) |
| -2/3 | 0.3490(1) | 0.3482(1) |
| -5/8 | 0.3521(5) | 0.3495(1) |
| -21/40 | 0.357(1) | 0.355(9) |

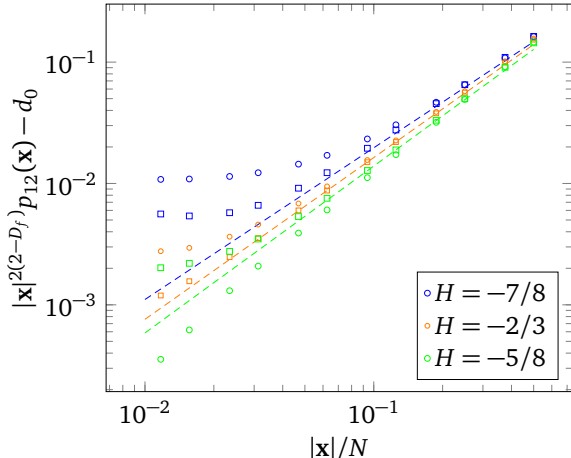

Figure 9: Numerical data for $|\mathbf{x}|^{2(2-D_f)}p_{12}(\mathbf{x})-d_0$ for $H=-7/8,-2/3,-21/40$, from surfaces (48) (circles) and (52) (squares). The lines show the prediction (27) with the exponent $2-1/\nu(H)$ given by (7).

## 3.2 Evidences of conformal invariance

With $M \neq N$, and following prediction (28), the quantity $\log\left[|\mathbf{x}|^{2(2-D_f^{(2)})}\left(p_{12}(\mathbf{x})-p_{12}(\mathbf{x}^\perp)\right)\right]$ should follow a line of slope 2. This is very clear for $H < -1/2$, as shown in Figure 10. When $H > -1/2$, the slope increases significantly: either there is no order 2 term (conformal invariance is broken), or this term is still present, with higher-order corrections making the effective slope significantly greater than 2. Assuming the latter and that the difference of connectivities is described by (29) on the whole line $H < 0$, we fit our data for different angles $\theta$ to the form:

$$|\mathbf{x}|^{2(2-D_f^{(2)})}\left(p_{12}(\mathbf{x})-p_{12}(\mathbf{x}^\perp)\right) = c_2(\theta)\left(\frac{|\mathbf{x}|}{N}\right)^2 + c_6(\theta)\left(\frac{|\mathbf{x}|}{N}\right)^6. \tag{33}$$

This fit shows good consistency with the data for all values of $H$, and allows to determine $c_2(\theta)$ with good precision. In Figure 11 we show that $c_2(\theta)$ has the expected behaviour (18): $c_2(\theta) \propto \cos(2\theta)$. This makes manifest the presence of a field with conformal dimension 2 and spin 2, and therefore of conformal invariance for all $H < 0$.

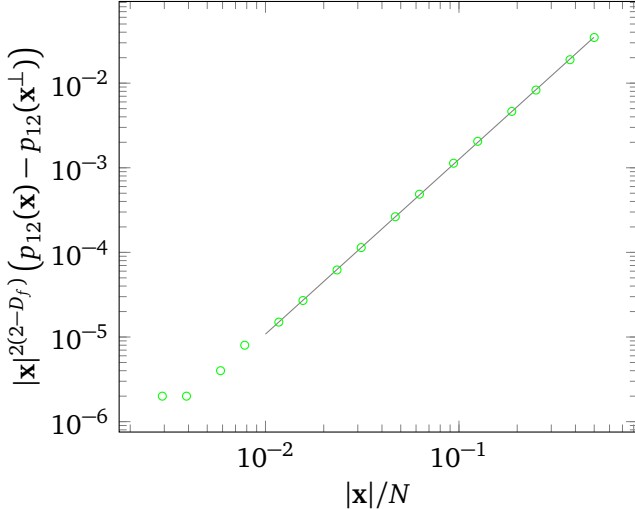

Figure 10: Difference of connectivities (28) for $H = -2/3$, measured for $M/N = 2$, $N = 2^{11}$ and $\theta = 0$. The best fit line has slope $\sim 2.07$, indicating the presence of the stress-energy tensor.

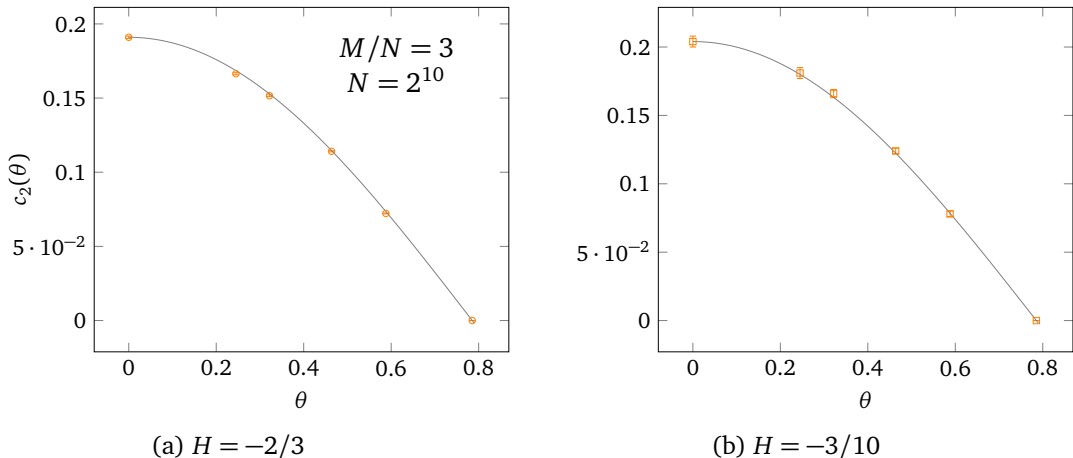

(a) $H = -2/3$

(b) $H = -3/10$

Figure 11: Values of $c_2(\theta)$ from fit (33), for different angles $\theta$, for $H < -1/2$ (a) and $H > -1/2$ (b). The curves show the prediction $c_2(\theta) = c_2(0)\cos(2\theta)$.

The behaviour of the order 6 coefficient is also in fair agreement with prediction (29), as shown in Figure 12.

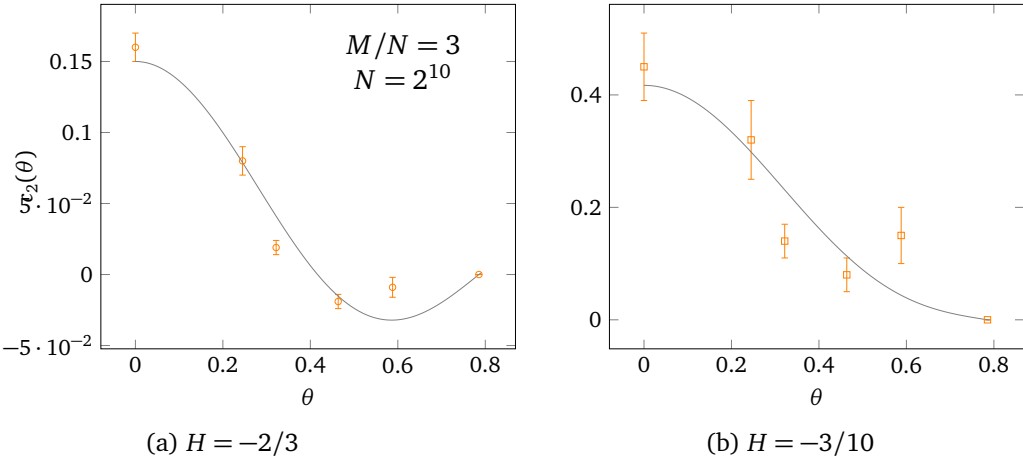

(a) $H = -2/3$             (b) $H = -3/10$

Figure 12: Values of $c_6(\theta)$ from fit (33), for different angles $\theta$, for $H < -1/2$ (a) and $H > -1/2$ (b). The curves are fits to the form (29): $c_6(\theta) = c_{6,2}\cos(2\theta) + c_{6,6}\cos(6\theta)$.

### 3.3 Spectrum and structure constants

Setting $\theta$ to zero, we varied the aspect ratio and obtained $c_v$ and $c_T$ as functions of $M/N$, given in Tables 2 and 5.

The coefficient $c_v$ is obtained by fitting the sum of connectivities $\frac{1}{2}|\mathbf{x}|^{2(2-D_f^{(2)})}\left(p_{12}(\mathbf{x}) + p_{12}(\mathbf{x}^\perp)\right)$ to the form (32). Taking the sum allows to remove the order 2 contributions of the stress-tensor fields.

Table 2: Best fit parameter $c_v(M/N)$, for different aspect ratios $M/N$. These values were obtained with the surface (52), which showed better convergence. When $H > -1/2$, the non-universal effects are too strong and are not described by the fit (32).

| $M/N$ \\ $H$ | 1 | 2 | 3 | 4 |
|---|---|---|---|---|
| percolation | 0.355402 | 0.185569 | 0.0964413 | 0.0501208 |
| -7/8 | 0.371(5) | 0.170(5) | 0.13(1) | 0.040(5) |
| -2/3 | 0.352(4) | 0.22(2) | 0.135(5) | 0.090(5) |
| -5/8 | 0.327(3) | 0.15(1) | 0.130(5) | 0.075(5) |

Figure 13 shows that the behaviour of $c_v(q)$ is in fair agreement with prediction (21):

$$c_v(q) \sim q^x, \tag{34}$$

with the slope $x$ given by the dimension of the spin field $x = 2\Delta_\sigma = 2 - D_f$, see Table 3. We point out that this behaviour is incompatible with the fact that the energy field is degenerate at level 2. Indeed, if it was degenerate the slope $x = 2\Delta_\sigma$ would be a continuously varying function of the central charge [50] and would be expected to show significant variation with $H$. In general, the presence of degenerate fields is a crucial feature of a CFT [64], which in some cases allow to solve the theory [65–68]. For pure percolation, the energy field is degenerate, which leads to relations between the different structure constants of the theory [43, 68, 69].

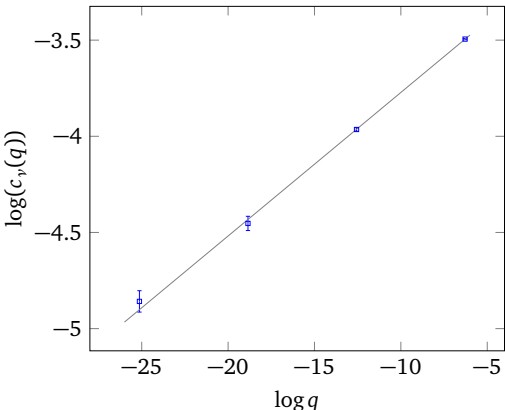

Figure 13: $c_\gamma$ as a function of $q$ and the best fit line, for $H = -2/3$.

Table 3: Exponent $x$ determining the behaviour of $c_\gamma(q)$ with $q$ (34), obtained from fitting $\log c_\gamma(q)$. These values are to be compared to the value of the spin dimension, which remains equal to the pure percolation value $2 - D_f^{\text{pure}} \sim 0.104$ when $H < -1/2$.

| $H$ | $x$ |
|---|---|
| -7/8 | 0.10(1) |
| -2/3 | 0.08(2) |
| -5/8 | 0.08(1) |

Setting $x$ to $2 - D_f$, a fit of $c_\gamma(q)$ as a function of $q^{2-D_f}$ gives an estimation of the quantity $\left[ C_{\sigma,\sigma}^{\varepsilon} \right]^2 n_\sigma$ (see 22), given in Table 4.

Table 4: Estimation of the coefficient $\left[ C_{\sigma,\sigma}^{\varepsilon} \right]^2 n_\sigma$. The percolation prediction was computed in [50].

| $H$ | $\left[ C_{\sigma,\sigma}^{\varepsilon} \right]^2 n_\sigma$ |
|---|---|
| pure percolation | $\pi\sqrt{3} \left( \frac{4}{9} \frac{\Gamma(7/4)}{\Gamma(1/4)} \right)^2 \sim 0.069$ |
| -7/8 | 0.07(1) |
| -2/3 | 0.05(1) |
| -5/8 | 0.04(1) |

Conversely, to obtain $c_T(q)$ we fit the difference $|\mathbf{x}|^{2(2-D_f^{(2)})} \left( p_{12}(\mathbf{x}) - p_{12}(\mathbf{x}^\perp) \right)$ to the form:

$$|\mathbf{x}|^{2(2-D_f^{(2)})} \left( p_{12}(\mathbf{x}) - p_{12}(\mathbf{x}^\perp) \right) = c_2(q) \left( \frac{|\mathbf{x}|}{N} \right)^2 + c_6(q) \left( \frac{|\mathbf{x}|}{N} \right)^6, \tag{35}$$

where

$$c_2(q) = 4d_0 \, c_T(q). \tag{36}$$

The values we obtained for $c_T(q)$, for both types of surfaces, are given in Tables 5, 6. Figure 14 shows the consistency betwwen the two sets of values, as expected from universality.

Table 5: Best fit parameter $c_2(M/N)/d_0$ for different aspect ratios $M/N$, for surfaces (48). The first line gives the numerical value of prediction (25) for pure percolation.

| $H$ ╲ $M/N$ | 1 | 2 | 3 | 4 | 5 |
|---|---|---|---|---|---|
| pure percolation | 0 | 0.3496 | 0.5109 | 0.5947 | 0.6383 |
| -7/8 | 0 | 0.376(5) | 0.531(5) | 0.610(5) | 0.645(5) |
| -2/3 | 0 | 0.383(5) | 0.547(5) | 0.607(5) | 0.640(5) |
| -5/8 | 0 | 0.395(5) | 0.555(5 ) | 0.619(5) | 0.641(5) |

Table 6: Best fit parameter $c_2(M/N)/d_0$ for different aspect ratios $M/N$, for surfaces (52).

| $H$ ╲ $M/N$ | 1 | 2 | 3 | 4 | 5 |
|---|---|---|---|---|---|
| -7/8 | 0 | 0.355(5) | 0.493(5) | 0.596(5) | 0.602(5) |
| -2/3 | 0 | 0.340(5) | 0.494(5) | 0.574(5) | 0.600(5) |
| -5/8 | 0 | 0.363(5) | 0.494(5) | 0.581(5) | 0.613(5) |

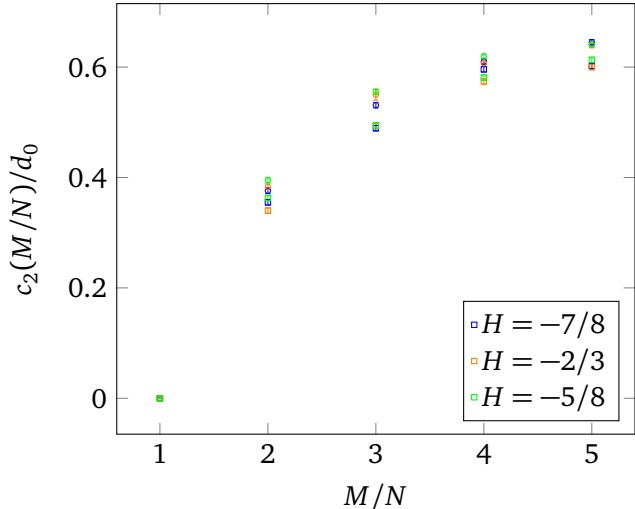

Figure 14: Comparison of the numerical values obtained for the universal quantity $c_2(M/N)/d_0$, for different Hurst exponents, for surfaces (48) (circles) and (52) (squares).

Following prediction (24), we fit the quantity $\log\left(2^{\frac{2-D_f}{3}}\pi^2 - \frac{c_2(q)}{d_0}\right)$ as a function of $\log q$ to a line. This is shown in Figure 15, and we obtain values for the dominant dimension $\Delta$ close to the dimension of the spin field, see Table 7.

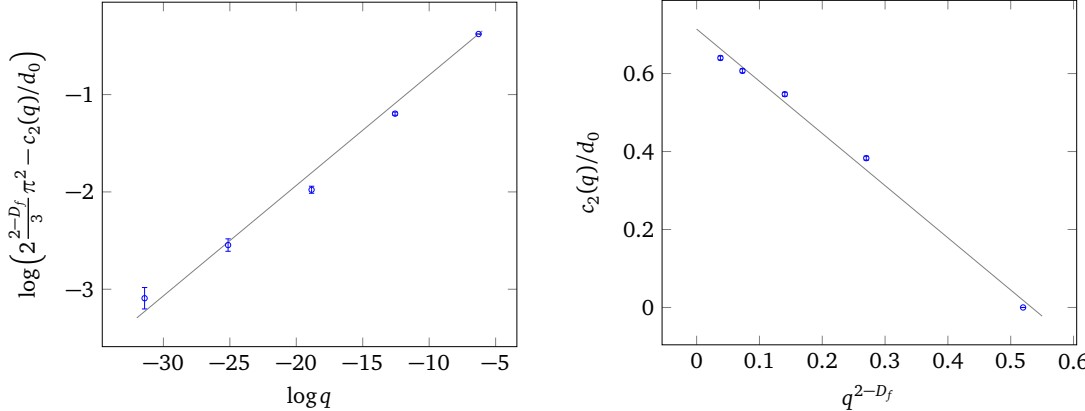

Figure 15: Numerical values at $H = -2/3$, for the quantities $\log\left(2\frac{2-D_f}{3}\pi^2 - c_2(q)/d_0\right)$ (left) and $c_2(q)/d_0$ (right), together with the corresponding best fit lines.

Table 7: Values of the dimension $2\Delta$ of the most dominant field obtained from fitting $\log\left(2\frac{2-D_f}{3}\pi^2 - \frac{c_2(q)}{d_0}\right)$, (1) for surfaces (48) and (2) for surfaces (52).

| $H$ | $2\Delta^{(1)}$ | $2\Delta^{(2)}$ |
|---|---|---|
| -7/8 | 0.12(1) | 0.10(1) |
| -2/3 | 0.11(1) | 0.09(1) |
| -5/8 | 0.12(1) | 0.10(1) |

Assuming that this dimension is indeed the one of the spin field, $2\Delta = 2\Delta_\sigma = 2 - D_f$, we fit $c_2(q)/d_0$ as a function of $q^{2-D_f}$:

$$c_2(q)/d_0 = c_2(0)/d_0 + a\,y, \quad y = q^{2-D_f}, \tag{37}$$

see Figure 15. In particular, from (24):

$$\frac{1}{12(2-D_f)}\frac{a}{c_2(0)/d_0} = \frac{n_\sigma}{c}. \tag{38}$$

The values of the cylinder ($q \to 0$) limit and of the ratio $n_\sigma/c$ obtained are given in Tables 8 and 9.

Table 8: Cylinder limit $c_2(0)/d_0$ and ratio of the spin field multiplicity $n_\sigma$ to the central charge $c$, obtained from fit (37), for surfaces (48).

| $H$ | $c_2(0)/d_0$ | $n_\sigma/c$ |
|---|---|---|
| pure percolation | $\frac{2(2-D_f)\pi^2}{3} \sim 0.6854$ | $\frac{4\pi}{5\sqrt{3}} \sim 1.4510$ |
| -7/8 | 0.71(2) | 1.51(7) |
| -2/3 | 0.71(2) | 1.50(9) |
| -5/8 | 0.72(2) | 1.5(1) |

Table 9: Cylinder limit $c_2(0)/d_0$ and ratio of the spin field multiplicity $n_\sigma$ to the central charge $c$, obtained from fit (37), for surfaces (52).

| $H$ | $c_2(0)/d_0$ | $n/c$ |
|---|---|---|
| -7/8 | 0.67(2) | 1.51(8) |
| -2/3 | 0.66(2) | 1.52(5) |
| -5/8 | 0.68(2) | 1.51(7) |

When $H > -1/2$, we could not determine the value of the plateau $d_0$, so we cannot determine the leading dimension in the expansion (24) as above. In Figure 16 we show the behaviour of $c_2(q)$ with $q^{2-D_f(H)}$, with $D_f(H)$ from Table 12. The points corresponding to large $M/N$ deviate significantly from a line. This could be explained by the fact that, when $H \to 0$, the fractal dimension $D_f \to 2$, so that the coefficient of the $q^{2-D_f}$ term in (24) becomes small and subleading terms in this expansion become non-negligeable.

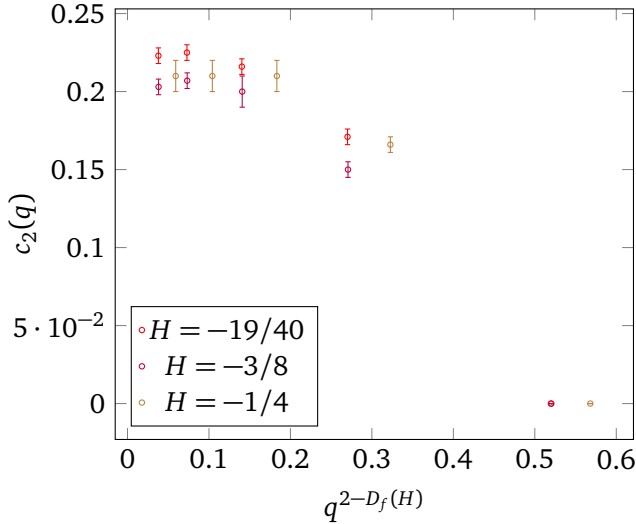

Figure 16: Behaviour of the coefficient $c_2(q)$ in the range $H > -1/2$.

## 4 Conclusion

In this paper we have studied the percolative properties of fractional random surfaces with negative Hurst exponent $H$. Via the connected components of their excursion sets, the level clusters, this problem is reformulated in terms of a long-range correlated two-dimensional site percolation model. The main motivation here was to better understand the universality of their percolation critical points, in particular in the region $-3/4 < H < 0$ where the correlation effects drive the system into universality classes different from the one of pure percolation. When the problem is defined on a rectangular domain of size $M \times N$ with toric boundary conditions, we argued that the two-point connectivity (10) represents an excellent observable to test conformal invariance. On the basis of three main assumptions, explained in Section 2.4, we predicted the leading contributions to the toric corrections, see (14) and (29). We tested these predictions by generating two types of fractional random surfaces (48) and (52), expected to have the same long distances behaviour. The comparison between the theory and the numerical simulations is summarised in Section 2.6. The main result is shown in Figure 10 and in Figure 11 and points out, for the first time, the existence of the two components of a traceless stress-energy tensor for all $H < 0$. Furthermore, the two point connectivity

on rectangular torus lattices gives access to the spectrum and to some fundamental structure constants of the underlying CFT, still unknown for any $H < 0$. Importantly, we find that the energy field in this CFT cannot be degenerate, whereas this is the case for pure percolation. We show that the leading contribution to the conformal partition function is the magnetic field $\sigma$ with scaling dimension $2 - D_f$, as shown in Figure 15 and in Table 7. The ratio $n_\sigma / c$ of the multiplicity of the magnetic field to the central charge has also been determined numerically with quite good precision, and it is reported in Table 8. Finally, we succeeded in evaluating the product $\left[ C_{\sigma,\sigma}^\varepsilon \right]^2 n_\sigma$, directly proportional to the fusion between the thermal and magnetic field. The results are given in Table 4. We conclude by noting that the fact that, for $H < -3/4$, the long-range correlation is irrelevant is a very established one. Nevertheless, the results in Table 4 verify this conjecture at the level of the structure constants of the theory, which encode much more information than the critical exponents. At the best of our knowledge, this is the first time such verification has been done. A last noteworthy observation concerns the corrections to the scaling of the critical level, when using the Binder method to locate the critical point (see Appendix B). From the values of the corresponding exponent $\omega$ given in Table 11, we argue that the long-range correlations break the integrability of the model.

## Acknowledgements

We thank Marco Picco for explaining us many crucial aspects on the numerical analysis of critical percolation points, and Hugo Vanneuville for sharing his insights and guiding us through the mathematical literature. We thank also Sylvain Ribault and Hans Herrmann for useful discussions. SG acknowledges support from a SENESCYT fellowship from the Government of Ecuador as well as from CNRS in the last part of the project.

## A   Fractional Gaussian surfaces

To generate a random function $u(\mathbf{x})$ satisfying the properties (1), we use a method based on the Fourier Filtering Method [9]. The principle is to create correlated random variables by linearly combining uncorrelated ones. Let us first briefly sketch the method. Given a set of uncorrelated random variable $w(\mathbf{x})$, $\mathbb{E}\left[ w(\mathbf{x})w(\mathbf{y}) \right] = \delta_{\mathbf{x},\mathbf{y}}$, one can define, via a convolution, a new set of random variables $u(\mathbf{x})$:

$$u(\mathbf{x}) = \sum_{\mathbf{y}} S(\mathbf{x} - \mathbf{y})w(\mathbf{y}). \tag{39}$$

The convolution kernel $S(\mathbf{x})$ is a non-random function which determines the $u(\mathbf{x})$ covariance function:

$$\mathbb{E}\left[ u(\mathbf{x})u(\mathbf{y}) \right] = \sum_{\mathbf{z}} S(\mathbf{x} - \mathbf{z})S(\mathbf{y} - \mathbf{z}). \tag{40}$$

By Fourier transforming both sides of the above equation, one can see that the large distance asymptotics (1) is determined by the small $\mathbf{k}$ asymptotics of $\hat{S}(\mathbf{k})^2$, where $\hat{S}(\mathbf{k})$ is the Fourier transform of $S(\mathbf{x})$. In particular, $\hat{S}(\mathbf{k}) \sim |\mathbf{k}|^{-H-1}$ (for $|\mathbf{k}| << 1$).

We apply this procedure to generate random long-range correlated surfaces. We consider a domain $[0, \cdots, N-1] \times [0, \cdots, M-1] \subset \mathbb{Z}^2$ where $\mathbf{x} = (x_1, x_2)$ denotes a lattice site:

$$\mathbf{x} = (x_1, x_2), \quad x_1 = 0, \cdots N-1$$

and

$$x_2 = 0, \cdots, M-1. \tag{41}$$

A random function $w(\mathbf{x})$ is generated by drawing its values independently at each point by an initial Gaussian distribution $P(w) = \mathcal{N}(0,1)$. The probability distribution function $P[w(\mathbf{x})]$ is therefore:

$$P[w(\mathbf{x})] = \prod_{\mathbf{x}} \frac{e^{-\frac{w(\mathbf{x})^2}{2}}}{\sqrt{2\pi}}. \tag{42}$$

The discrete Fourier transform of $w(\mathbf{x})$ is defined as:

$$\hat{w}(\mathbf{k}) = \frac{1}{NM} \sum_{\mathbf{x}} w(\mathbf{x}) e^{-i\mathbf{k}\mathbf{x}} = \frac{1}{NM} \sum_{x_1=0}^{N-1} \sum_{x_2=0}^{M-1} w(x_1, x_2) e^{-2\pi i \left(x_1 \frac{k_1}{N} + x_2 \frac{k_2}{M}\right)}, \tag{43}$$

where

$$\mathbf{k} = 2\pi \left(\frac{k_1}{N}, \frac{k_2}{M}\right), \quad k_1 = 0, \cdots, N-1, \; k_2 = 0, \cdots, M-1. \tag{44}$$

From (42) one has:

$$\mathbb{E}[\hat{w}(\mathbf{k})] = 0, \quad \mathbb{E}[\hat{w}(\mathbf{k})\hat{w}(\mathbf{p})] = \delta_{k_1, N-p_1} \delta_{k_2, M-p_2}. \tag{45}$$

We use the convolution kernel:

$$\hat{S}(\mathbf{k}) = \begin{cases} = \lambda_{\mathbf{k}}^{-\frac{H+1}{2}}, & \text{for } k_1, k_2 \neq 0 \\ = 1 & \text{for } k_1 = k_2 = 0, \end{cases} \tag{46}$$

where:

$$\lambda_{\mathbf{k}} = \left(2\cos\left(\frac{2\pi}{N} k_1\right) + 2\cos\left(\frac{2\pi}{M} k_2\right) - 4\right). \tag{47}$$

We generate the random surface $u(\mathbf{x})$ by doing the following inverse Fourier transform:

$$u(\mathbf{x}) = \frac{1}{\text{norm}} \sum_{\mathbf{k}} \hat{S}(\mathbf{k})\, \hat{w}(\mathbf{k})\, e^{i\mathbf{k}\mathbf{x}}, \quad \text{norm} = \sum_{\mathbf{k}} \hat{S}^2(\mathbf{k}). \tag{48}$$

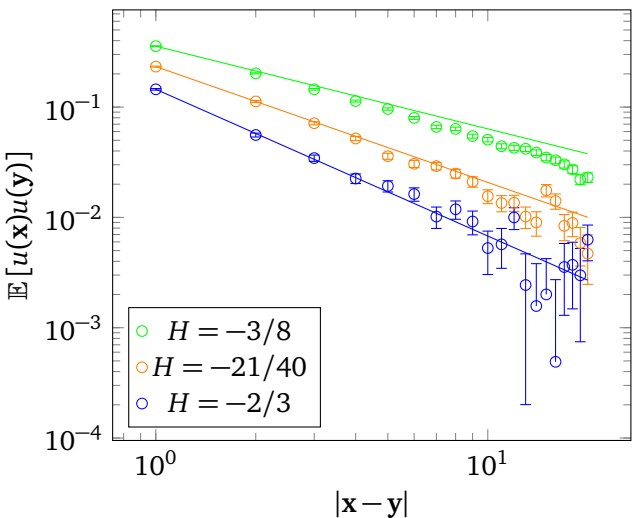

Figure 17: Numerical measurement of $\mathbb{E}[u(\mathbf{x})u(\mathbf{y})]$ for different values of the Hurst exponent, on square lattices of size $M = N = 2^8$. The lines have slopes $-2H$.

The universal properties do not depend on the initial distribution $P[w(\mathbf{x})]$ distribution nor on the precise form of the kernel as long as $\hat{S}(\mathbf{k})$ has the same small $\mathbf{k}$ asymptotic behaviour

[70]. As we explain in Section 3, we find useful to generate long-range correlated random surfaces by using another distribution $P_2[w(\mathbf{x})]$ for $w(\mathbf{x})$ and a different kernel. In particular, the $P_2[w(\mathbf{x})]$ is determined by the uniform distribution:

$$P_2[w(\mathbf{x})] = \prod_{\mathbf{x}} P(w(\mathbf{x})), \quad P(w(\mathbf{x})) = \left\{ \begin{array}{ll} 1, & |w(\mathbf{x})| < \frac{\sqrt{3}}{N} \\ 0, & |w(\mathbf{x})| > \frac{\sqrt{3}}{N} \end{array} \right. \tag{49}$$

and the kernel:

$$\hat{S}_2(\mathbf{k}) = \left\{ \begin{array}{ll} |\mathbf{k}|^{-H-1} & \text{for } \mathbf{k} \neq (0,0), \\ 1 & \text{for } \mathbf{k} = (0,0) \end{array} \right. , \tag{50}$$

where:

$$|\mathbf{k}| = \frac{2\pi}{N}\sqrt{k_1^2 + k_2^2}, \quad k_1, k_2 = -N/2, \cdots N/2 - 1. \tag{51}$$

The second kind of surfaces we generate are

$$u(\mathbf{x}) = \frac{1}{\text{norm}} \sum_{\mathbf{k}} \hat{S}_2(\mathbf{k})\, \hat{w}_2(\mathbf{k})\, e^{i\,\mathbf{k}\,\mathbf{x}}, \quad \text{norm} = \sum_{\mathbf{k}} \hat{S}_2^2(\mathbf{k}), \tag{52}$$

where we indicated as $\hat{w}_2(\mathbf{k})$ the Fourier transforms of the random function $w(\mathbf{x})$ of law (49). In the above equations we assumed $M = N$, but the generalization to $M \neq N$ is straightforward. Note that, due to the (Lyupanov) central limit theorem, $\hat{w}_2(\mathbf{k})$ is described in the large $N$ limit by a Gaussian distribution and the function $u(\mathbf{x})$ can be considered an instance of a fractional Gaussian surface. For $H < 0$, the surface $u(\mathbf{x})$, generated by (48) or by (52):

- is real, $u(\mathbf{x}) \in \mathbb{R}$, from the property (45) and the symmetry of the kernel (46)

- satisfies (1). In Figure 17 we show the numerical measurements of $\mathbb{E}[u(\mathbf{x})u(\mathbf{y})]$ for the surface (48)and for different values of the roughness exponent. The data points are compared to the power law decay $|\mathbf{x} - \mathbf{y}|^{2H}$.

- has a zero mode which vanishes in law:

$$\mathbb{E}[\hat{u}(\mathbf{0})] = 0. \tag{53}$$

- is normalised such that:

$$\mathbb{E}[u(\mathbf{x})^2] = 1. \tag{54}$$

  Note that, in the thermodynamic limit, the normalisation constant in (48) is finite for negative $H$, as norm $\sim N^{2H} + O(1)$ ($N \gg 1, M/N = O(1)$). The surface fluctuations are thus bounded.

- satisfies periodic boundary conditions in both directions

$$u(\mathbf{x} + \mathbf{t}) = u(\mathbf{x}), \quad \text{for } \mathbf{t} = (n\,N, m\,M),\, n, m \in \mathbb{N}. \tag{55}$$

# B  Percolation phase transition: critical level $h_c$ and the critical exponents $\nu$ and $D_f$

We study here the critical percolative properties of the level clusters of the surface (48) and (52). In particular we determine numerically the critical level $h_c$ and the exponents $\nu$ and $D_f$.

## B.1 Critical level and correlation length exponent $\nu$

For a sign-symmetric random function $u(\mathbf{x})$ on the Euclidean space, $\mathbf{x} \in \mathbb{R}^2$, the critical level is $h_c = 0$ by symmetry argument [13]. Our function $u(\mathbf{x})$ is defined on a lattice and $h_c$ is expected to be negative. We determine the critical level $h_c$ by the standard procedure of percolation theory [11]. We consider square domains of different sizes $N \times N$. We determine the average $\mathbb{E}[h_c(N)]$ of the level $h_c(N)$ at which a level cluster connecting the top and the bottom of the lattice appears. This quantity scales with the size of the lattice as:

$$\mathbb{E}[h_c(N)] - h_c \sim N^{-\frac{1}{\nu}}. \tag{56}$$

The data point for $\mathbb{E}[h_c(N)]$, shown in Figure 18 as a function of $N^H$ for different values of $H$, are very well described by a linear interpolation, thus confirming the predition (7). Fitting the data to the form (56) with $\nu = \nu^{\text{long}}$, we obtain the values of $h_c$ reported in Table 10.

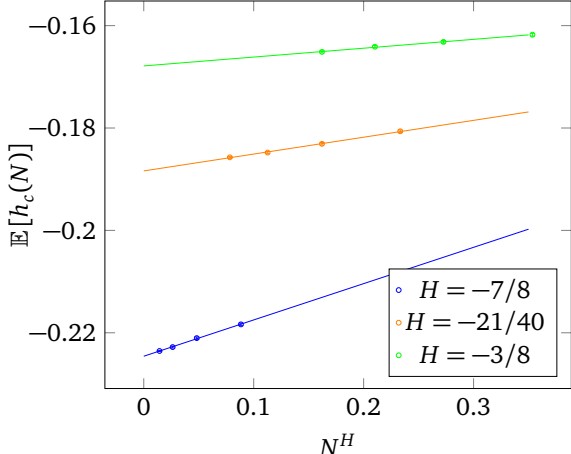

Figure 18: $\mathbb{E}[h_c(N)]$ for $N = 2^4, \cdots 2^7$ as a function of $N^H$. The lines are the best fits to the form (56) with $\nu = -1/H$ for different $H$s. The intercepts with the vertical $N^H = 0$ axis ($N \to \infty$ limit), give the estimation for $h_c$.

Table 10: Critical level obtained from scaling (56), for the surfaces (48).

| $H$ | $h_c$ |
|---|---|
| -7/8 | -0.2238(1) |
| -2/3 | -0.2034(1) |
| -5/8 | -0.1985(1) |
| -21/40 | -0.1860(2) |
| -19/40 | -0.1775(3) |
| -3/8 | -0.1670(5) |
| -3/10 | -0.1570(5) |

Another way to determine the critical point is based on the Binder method. We apply this method to study the surface (52). Defining the moments $M_m$ as:

$$M_m = \sum_{i=0}^{\infty} i^m n_i, \tag{57}$$

with $n_i$ the number of level clusters composed of $i$ sites, one computes the ratio $r_N^{\text{Bin}}(h)$

$$r_N^{\text{Bin}}(h) = \frac{\mathbb{E}\left[M_4\right]}{\mathbb{E}\left[M_2\right]^2}, \tag{58}$$

where the average $\mathbb{E}[\cdots]$ is weighted by the distribution (49). The ratio $r_N^{\text{Bin}}(h)$ depends on the level $h$ and on the system size $N$ through a scaling relation of the type:

$$r_N^{\text{Bin}}(h) = f\left((h-h_c)N^{\frac{1}{\nu}}\right) + a\,N^{-\omega}, \tag{59}$$

where the function $f$ is some scaling function, and the term $a\,N^{-\omega}$, with $a$ a non-universal prefactor, is a correction to the scaling term. The interpretation of $\omega$ is discussed below. From (59), one can find the point $h_c(N)$ where the curves $r_N^{\text{Bin}}(h)$ and $r_{2N}^{\text{Bin}}(h)$ intersect [71] and use the fitting form:

$$h_c(N) = h_c + \frac{a}{N^x}, \tag{60}$$

to determine $h_c$, with $x$ a free parameter. For each value in (30), we compute (58) for sizes $N = 2^s$, $s = 4, \cdots, 9$ and $N = 3 \times 2^s$, $s = 3, \cdots, 7$ averaged over $10^5$ instances. We interpolate the curves and find their intersections. The Binder method shows less precision for $H$ approaching 0. Indeed the correlation length exponent $\nu = -1/H$ increases fast, making the size effects much smaller. The curves $r_N^{\text{Bin}}(h)$ and $r_N^{\text{Bin}}(h)$ tend to be parallel, and localising their crossing point becomes difficult. In Figure 19 we show the scaling of the crossing points $h_c(N)$ for some values of $H$. Once the critical point is located, the thermal exponent $\nu$ can be estimated by using that:

$$\frac{d}{dh}r_N^{\text{Bin}}(h)|_{h=h_c} \sim N^{1/\nu}. \tag{61}$$

In Table 11 we give the values of $h_c$ obtained from (60), and the values of $\nu$ obtained from (61). These latter are in fair agreement with the prediction (6, 7). Setting $\nu$ to (7) we estimate the values of $\omega$ as $\omega = x - 1/\nu$.

Table 11: Values of the critical level $h_c$ obtained with the Binder method. The $\nu$ exponent is obtained from equation (61), and the value of the exponent $w$ is obtained from scaling (60), with $\nu$ set to (7). The measurements have been taken for the surface (52).

| $H$ | $h_c$ | $\nu$ | $\omega$ |
|---|---|---|---|
| -1 | -0.3210(9) | 1.33(2) | 2.00(5) |
| -7/8 | -0.3075(5) | 1.46(8) | 1.00(5) |
| -2/3 | -0.2793(5) | 1.67(5) | 0.8(1) |
| -5/8 | -0.2722(5) | 1.9(1) | 1.0(1) |

It is quite interesting to comment on the exponent $\omega$, which determines the correction to the scaling. The exponent $\omega$ is expected to be the conformal dimension of the first irrelevant thermal field. In [72] is was observed that, when the model is integrable, the corrections to the scaling are always associated to irrelevant fields that appear in the fusion between relevant ones. To be more specific, the authors of [72] considered those statistical models that are described by rational CFTs. The spectrum of these CFTs contain a finite set of primary fields, which close under Operator Product Algebra and which are listed in the so-called Kac Table. When these models are integrable, the correction to scaling are therefore determined by fields inside the Kac table. In the pure percolation CFT, the (relevant) energy density $\varepsilon$ field, $\varepsilon = V_{1,2}$ generate by fusion with itself an infinite series of irrelevant fields with dimension

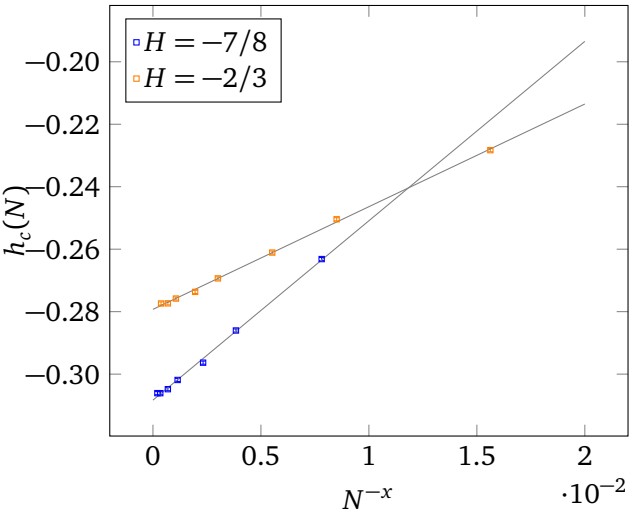

Figure 19: Values of $h_c(N)$ obtained from the crossing of the curves $r_N^{\text{Bin}}(h)$ and $r_N^{\text{Bin}}(h)$, defined in (58). Measurements have been taken for the surface (52).

$\Delta_{1,n}$, $n = 3, 4, ...$ (note that we have used the standard minimal model notation $V_{r,s}$ and $\Delta_{r,s}$ for the field and conformal dimension). In the case of pure percolation, which is an integrable model, the value of $\omega$ is therefore expected to be given by the lowest irrelevant thermal field dimension, $\omega = 2\Delta_{1,3} = 2$. A discussion of this exponent can be found for instance in Appendix D of [73]. In the case of pure percolation, we find indeed $\omega = 2$. We observe in Table 11 that, when $H \neq -1$, a non-universal correction with $\omega \sim 1$ to the scaling dominates. v

## B.2 Fractal dimension $D_f$

At the critical point $h = h_c$, the level clusters have fractal dimension $D_f$. This dimension determines the scaling of the average mass (i.e. number of points) $\mathcal{A}_l$ of a level cluster with respect to its length $l$, $\mathcal{A}_l \sim l^{D_f}$. The length of a level cluster can be defined as its radius of gyration. One effective way to measure $D_f$ is to consider the percolating level cluster whose size is of the same order of the system size, $l \sim N$. To determine $D_f$, we use then the following relation:

$$\mathbb{E}[\# \text{ sites of the p.l.c.}] \sim N^{D_f}, \quad \text{p.l.c.=percolating level cluster.} \tag{62}$$

A representative example of a numerical measurement of the above average is shown in Figure 20a, for $H = -2/3$. To remove the small sizes effects, we perform fits with the successive lower sizes removed, and expect the best fit parameter to converge to the fractal dimension, as in Figure 20b. The values $D_f^{(1)}$ obtained are given in Table 12.

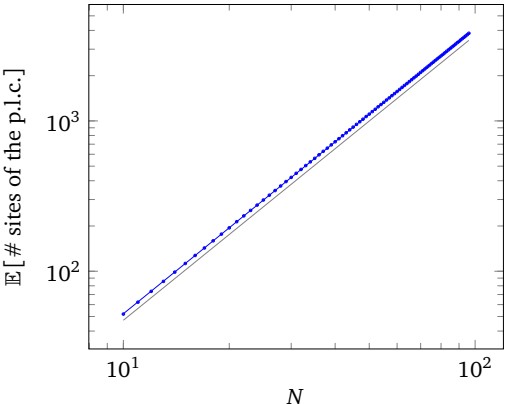

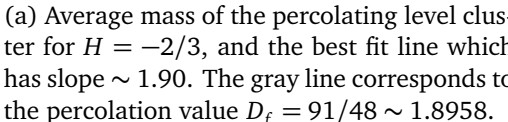

(a) Average mass of the percolating level cluster for $H = -2/3$, and the best fit line which has slope $\sim 1.90$. The gray line corresponds to the percolation value $D_f = 91/48 \sim 1.8958$.

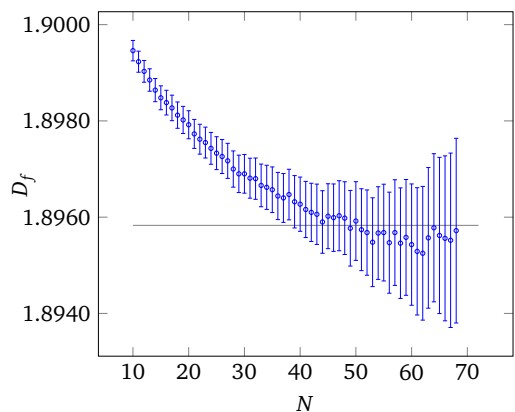

(b) Convergence of the best fit parameter for the fractal dimension when the lowest size points are removed. Here it converges to the percolation value shown as a grey line.

Figure 20

Table 12: Fractal dimensions obtained (1) from the scaling of the largest cluster (62) and (2) from the power-law decay of the two-point connectivity (11), and comparison with previous numerical work [25].

| $H$ | $D_f^{(1)}$ | $D_f^{(2)}$ | $D_f$ [25] |
|---|---|---|---|
| -7/8 | 1.8955(5) | 1.8945(2) | 1.8964(2) |
| -2/3 | 1.8960(10) | 1.893(1) | |
| -5/8 | 1.8955(6) | 1.892(1) | 1.8950(3) |
| -21/40 | 1.8965(10) | 1.8910(5) | |
| -19/40 | 1.8955(8) | 1.8897(5) | |
| -3/8 | 1.904(1) | 1.8970(5) | 1.9006(4) |
| -1/4 | 1.917(1) | 1.906(1) | 1.9128(5) |

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
