# Peer review of "Topological effects and conformal invariance in long-range correlated random surfaces"

_SciPost Physics, doi:SciPost Phys. 9, 050 (2020)_

## Round 1 · Referee Report · Anonymous (Referee 1) · 2020-7-17

Strengths

1- Investigates level sets of correlated random surfaces using an astute combination of CFT predictions and high-quality numerical simulations. 2- Establishes a range of Hurst exponents for which there is a continuous variation of the conformal data. 3- Applies methods of random conformal geometry to a new class of models.

Weaknesses

1- While generally well-written, there is often a long distance between the figures and tables, and the place in the text where they are discussed. This makes the paper more difficult to read. 2- Many of the items in the bibliography are not complete, making it unnecessarily difficult to identify the cited literature.

Report

The authors studies long-range correlated correlated random surfaces via their critical level sets, using methods of conformal random geometry. The main object studied is the two-point connectivity correlator of level clusters in the toroidal geometry.

Using the assumption of conformal invariance, and some technical assumptions on the field content, a number of precise predictions for the distance, angular and aspect ratio dependence of the two-point functions are established, building heavily on a previous paper (ref. [47]) by the same authors. These predictions are then checked by high-quality numerical simulations. The authors convincingly establish that for Hurst exponents in the range -3/4 < H < 0, there is a critical phase with continuously varying exponents and structure constants. However, the authors do not manage to establish analytically the values of these conformal data.

The paper is generally well-written and easy to follow for readers knowledgeable of conformal field theory. The conclusions are interesting, well argued and of importance for the model being studied. However, the structure in which the numerical results are concentrated in section 3, but discussed mainly in section 2, makes the jumping back and forth a bit annoying at times.

Another point that needs improvement is the bibliographic information: often published papers are only referred to by the name of the authors and the title (e.g. refs. [17, 24, 30, 51, 59]), sometimes a preprint number is provided but not the published journal version (e.g. refs. [16, 37]), and for some recent preprints not even the arXiv number is provided (e.g. refs. [39, 40, 47]).

In addition I have a number of minor comments:

1- Above (2.10) the "spectrum" S is defined to be all the fields present in the theory. Does (2.10), being a very specific correlation function, contain non-zero contributions from all fields, or only a suitably defined subset thereof? 2- On the first line of page 11, should p be p_{12}? 3- In the discussion about the lattice representation of the stress-energy tensor on page 12, the authors might wish to comment on W.M. Koo and H. Saleur, hep-th/9312156. 4- Around (2.21), are the coefficients c_{6,2} and c_{6,6} known in explicit form, like in (2.11)? 5- In the discussion about the angular dependence of spinfull fields below (3.4), the authors might wish to mention R. Couvreur et al, arXiv:1704.02186, as well as X. Tan et al, arXiv:1809.06650, where similar situations were considered.

Requested changes

1- Consider approaching figures and tables with the place in the text where they are discussed. 2- Provided the missing detailed bibliographical information. 3- Address the minor points mentioned in the report.

  • validity: high
  • significance: high
  • originality: high
  • clarity: high
  • formatting: excellent
  • grammar: excellent

Author:  Nina Javerzat  on 2020-08-04  [id 915]

(in reply to Report 1 on 2020-07-17)

We are grateful to the referee for carefully reading our paper and for her·his valuable report. In the following, we answer in detail to the referee’s points and outline the changes that have been made.

  1. Consider approaching figures and tables with the place in the text where they are discussed. Our paper presents theoretical predictions which are compared to numerical measurements. We have decided to separate the theoretical discussion (Section 2) from the numerical one (Section 3). We believe that moving the main figures to Section 2 would dilute the theoretical explanation, since it would be necessary to accompany them with a lot of details on the numerical analysis. Therefore we prefer to keep the actual structure of the paper, where Section 2.6 summarises the main theoretical predictions, and refers to the corresponding figures and tables of Section 3.

  2. Provide the missing detailed bibliographical information. We thank the referee for listing the missing bibliographic information. We have improved the bibliography; in particular we have provided the references to the published journal versions.

  3. Address the minor points mentioned in the report:

    1. Above (2.10) the "spectrum" S is defined to be all the fields present in the theory. Does (2.10), being a very specific correlation function, contain non-zero contributions from all fields, or only a suitably defined subset thereof? We wrote equation (18) (previously (2.10) ) in the most general form consistent with the symmetry between holomorphic and anti-holomorphic sectors. Indeed, one does not expect all the fields to contribute, since fields can have vanishing structure constant or one-point function. We have added a sentence after equation (18) to make this point clearer.

    2. On the first line of page 11, should p be p_{12}? Indeed it should, we have added the missing subscript.

    3. In the discussion about the lattice representation of the stress-energy tensor on page 12, the authors might wish to comment on W.M. Koo and H. Saleur, hep-th/9312156. We thank the referee for pointing out this relevant reference, that has been added as [61].

    4. Around (2.21), are the coefficients c_{6,2} and c_{6,6} known in explicit form, like in (2.11)? These coefficients are fixed by the Virasoro algebra, under the assumption that they get contributions only from the Identity descendants, and can therefore be obtained in explicit form. However this computation is cumbersome, and the numerical evaluation of these coefficients is not precise enough to make the comparison with the theoretical result meaningful. We have added a paragraph below equation (29) (previously (2.21)) to explain this point.

    5. In the discussion about the angular dependence of spinfull fields below (3.4), the authors might wish to mention R. Couvreur et al, arXiv:1704.02186, as well as X. Tan et al, arXiv:1809.06650, where similar situations were considered. We thank the referee for pointing out these references. We have added in the first paragraph of page 12 a phrase in which we refer the reader to these two references.

---

## Round 1 · Referee Report · Anonymous (Referee 2) · 2020-7-18

Strengths

1- very clearly written 2- introduces CFT concepts for non-experts 3- convincing evidence of conformal invariance

Weaknesses

1- lots of plots in section 3, from which it is difficult to evaluate where the most important information is (for instance, the dependence of the critical exponents in the parameter H) 2- not clear to me why Appendix B is not included in the main text, as it contains valuable information on the critical exponents

Report

The authors study a family of long-range percolation models, which are defined from the set of "active sites" in random surface models (sites whose height is larger than some fixed level $h$). The random surfaces are characterized by their Hurst exponent $H$, and the authors focus in particular on the line of critical points $- \frac34 < H < 0$, for which the critical exponents are unknown, and the existence of conformal symmetry even debated.
This paper brings significant progress in this direction : through studying numerically the probability that two points on a torus belong to the same level clusters, and making comparison with predictions from CFT (under some reasonable assumptions), the authors provide strong evidence for conformal invariance, as well as numerical estimates for some of the critical exponents, as well as structure constants.

This is a very nice and well-written paper, which I think deserves publication in SciPost after the issues listed in the following Section have been addressed.

Requested changes

1- a reader with some knowledge on CFT but less on geometrical models might be surprised at first sight that it is so difficult to understand the CFT at play here, given the known classification of 2D CFTs, Kac table, etc... If I understand correctly, the difficulty has to do with the non-locality of the considered object, which may also be reexpressed in terms of local observables in a non-unitary theory. Is this the point ? If so, it might be useful to add a few lines of explanation, either in the Intro, or in Section 2.3, or in Appendix B.

2- In relation with the above remark : the authors argue in Appendix B.1 that the exponent \omega should belong to the Kac table if the theory were integrable, and conclude from there about the non-integrability of the long-range percolation : is it really true ? Does the "Kac table argument" really hold in the non-unitary setting ?

3- maybe before eq (1.4) explain briefly the idea behind the construction of the surface, which is to generate it through convolutions of uncorrelated Gaussian variables

4- It would be useful to maybe give more detail on the numerics (for instance, was the data from Fig. 2.1 obtained by averaging over many different random samples ? How many? etc...)

5- it would be useful to add a standard reference for the scaling relations (2.8)

6- When introducing eq. (2.6), maybe worth recalling that in the particular case of percolation such a formula has been established in Ref [47]

7- Finally, a few typos (I might have spotted many more) : - legend of Fig. 2.1 : "according to Table B.6" : shouldn't it rather be Table B.2 ? -Sec. 2.3, second paragraph : "correspondance" -> "correspondence" -Before eq. 2.8 : "their dimensionS" -in the conclusion : "traceles stress-energy tensor" -> traceless

  • validity: high
  • significance: high
  • originality: high
  • clarity: top
  • formatting: excellent
  • grammar: excellent

Author:  Nina Javerzat  on 2020-08-04  [id 916]

(in reply to Report 2 on 2020-07-18)

We are grateful to the referee for carefully reading our paper and for her·his valuable report. In the following, we answer in detail to the referee's points and outline the changes that have been made.

Weaknesses

  1. Lots of plots in section 3, from which it is difficult to evaluate where the most important information is (for instance, the dependence of the critical exponents in the parameter H) Our numerical analysis of the model involved considering different types of surfaces, as well as different parameters (Hurst exponent, geometry of the domain). To justify our numerical approach and put our results on solid grounds we thought necessary to include these plots in the paper, gathered in Section 3. In Section 2.6 we highlight which figures and tables correspond to the main results.

  2. not clear to me why Appendix B is not included in the main text, as it contains valuable information on the critical exponents Initially the content of Appendix B was indeed in the main text, since the measurements presented are of interest as they support previous results and also introduce new observations (exponent \omega). However we decided to include it instead as an appendix because we wanted the reader to focus on what we think is really new in this paper, namely the study of the two-point connectivity and the manifestation of conformal invariance in the toroidal effects.

Requested changes

  1. a reader with some knowledge on CFT but less on geometrical models might be surprised at first sight that it is so difficult to understand the CFT at play here, given the known classification of 2D CFTs, Kac table, etc... If I understand correctly, the difficulty has to do with the non-locality of the considered object, which may also be reexpressed in terms of local observables in a non-unitary theory. Is this the point ? If so, it might be useful to add a few lines of explanation, either in the Intro, or in Section 2.3, or in Appendix B. Indeed the CFTs expected to describe geometrical models do not correspond to ANY known CFT. This is well illustrated by the simplest model of pure percolation, for which -- as the referee mentioned, the non-locality can be traded for a local non-unitary CFT with indecomposable but non irreducible Virasoro representations (logarithmic CFT). Local logarithmic CFTs have not been fully solved. Following the suggestion of the referee we have added a few lines of explanation in the second paragraph of page 5: "[...] Contrary to ... in this paper."

  2. In relation with the above remark : the authors argue in Appendix B.1 that the exponent \omega should belong to the Kac table if the theory were integrable, and conclude from there about the non-integrability of the long-range percolation : is it really true ? Does the "Kac table argument" really hold in the non-unitary setting ? We thank the referee for pointing out something that had indeed to be clarified. The existing literature on this issue (for instance Blote et al. [72]) generally considers (unitary) rational CFTs statistical models, in which a finite number of primary fields closes under the operator algebra. Their statement is that the correction to scaling comes, for integrable models, from irrelevant fields in the Kac table. This means that irrelevant fields are obtained by fusion of relevant ones. In the (non-unitary) non-rational setting, the operator algebra does not close under a finite number of primaries. However the statement that the correction to scaling is associated to irrelevant fields obtained by fusion of relevant ones still holds. We have explained this point below Table 11 in Appendix B "In [72] ... = 2".

  3. maybe before eq (1.4) explain briefly the idea behind the construction of the surface, which is to generate it through convolutions of uncorrelated Gaussian variables We have added a line of explanation below this equation (now equation (5) ).

  4. It would be useful to maybe give more detail on the numerics (for instance, was the data from Fig. 2.1 obtained by averaging over many different random samples ? How many? Etc…) In the captions of the two figures showing numerical data in section 2 (fig. 3 and 5), we added more details on how the data are taken. Moreover we refered the reader to Section 3 where he·she can find all the details on the numerical simulation (see also our reply to Requested change 1 in Report 1).

  5. it would be useful to add a standard reference for the scaling relations (2.8) We have added the relevant references (now [53, Chapter 3] and [11, Section 3.3]) before equation (16) (previously (2.8) ).

  6. When introducing eq. (2.6), maybe worth recalling that in the particular case of percolation such a formula has been established in Ref [47] We have added the reference to [50] (previously [47]) below equation (14) (previously (2.6) ).

---

## Round 2 · Referee Report · Anonymous (Referee 1) · 2020-8-5

Report

The authors have taken into account all my comments and I now recommend publication.

---

## Round 2 · Referee Report · Anonymous (Referee 2) · 2020-8-6

Report

I thank the authors for the modifications brought to their manuscript, which in my opinion make it ready for publication in SciPost.

---

## Editorial Decision

published